# mTORC1 and mTORC2 differentially promote natural killer cell development

Chao Yang[1,2], Shirng-Wern Tsaih[3,4], Angela Lemke[3,4], Michael J Flister[3,4], Monica S Thakar[1,5], Subramaniam Malarkannan[1,2,5,6]*

[1]Laboratory of Molecular Immunology and Immunotherapy, Blood Research Institute, Blood Center of Wisconsin, Milwaukee, United States; [2]Departments of Microbiology and Immunology, Medical College of Wisconsin, Milwaukee, United States; [3]Human and Molecular Genetics Center, Medical College of Wisconsin, Milwaukee, United States; [4]Departments of Physiology, Medical College of Wisconsin, Milwaukee, United States; [5]Departments of Pediatrics, Medical College of Wisconsin, Milwaukee, United States; [6]Departments of Medicine, Medical College of Wisconsin, Milwaukee, United States

**Abstract** Natural killer (NK) cells are innate lymphoid cells that are essential for innate and adaptive immunity. Mechanistic target of rapamycin (mTOR) is critical for NK cell development; however, the independent roles of mTORC1 or mTORC2 in regulating this process remain unknown. *Ncr1*[iCre]-mediated deletion of *Rptor* or *Rictor* in mice results in altered homeostatic NK cellularity and impaired development at distinct stages. The transition from the CD27$^+$CD11b$^-$ to the CD27$^+$CD11b$^+$ stage is impaired in *Rptor* cKO mice, while, the terminal maturation from the CD27$^+$CD11b$^+$ to the CD27$^-$CD11b$^+$ stage is compromised in *Rictor* cKO mice. Mechanistically, Raptor-deficiency renders substantial alteration of the gene expression profile including transcription factors governing early NK cell development. Comparatively, loss of Rictor causes more restricted transcriptome changes. The reduced expression of T-bet correlates with the terminal maturation defects and results from impaired mTORC2-Akt$^{S473}$-FoxO1 signaling. Collectively, our results reveal the divergent roles of mTORC1 and mTORC2 in NK cell development.

DOI: https://doi.org/10.7554/eLife.35619.001

*For correspondence: Subramaniam.Malarkannan@bcw.edu

**Competing interests:** The authors declare that no competing interests exist.

## Introduction

Natural killer (NK) cells are innate lymphocytes capable of mediating both cytotoxicity and cytokine production in response to transformed or virally infected cells (*Sun and Lanier, 2011*; *Vivier et al., 2008*). Humans with NK cell deficiency are more susceptible to viral infection, especially herpesviruses (*Orange, 2002*). Their natural killing potential makes NK cells an ideal candidate for immunotherapy against various tumors, and NK cell adoptive transfer has been explored as promising adjuvant therapy (*Curti et al., 2011*; *Miller et al., 2005*; *Rubnitz et al., 2010*). Given the importance of NK cells, it is critical to define the essential developmental programs that regulate their homeostasis and maturation.

NK cells develop in the bone marrow (BM) and progress through distinct differentiation stages (*Di Santo, 2006*). Expression of IL-15/IL-2 receptor β chain (CD122) defines the commitment of common lymphoid progenitor (CLP) cells to the NK cell lineage (*Rosmaraki et al., 2001*). Recently, the cell surface markers-defined NK cell progenitors (NKP) have been refined as Lin$^-$Flt3$^-$CD27$^+$2B4$^+$-CD127$^+$CD122$^+$NK1.1$^-$ cells (*Carotta et al., 2011*; *Fathman et al., 2011*). Committed NKPs differentiate into immature (iNK; Lin$^-$CD122$^+$NK1.1$^+$DX5$^-$) and eventually mature NK cells (mNK; Lin$^-$CD122$^+$NK1.1$^+$DX5$^+$) (*Di Santo, 2006*; *Kim et al., 2002b*; *Rosmaraki et al., 2001*;

*Williams et al., 2000*). Development of NK cells following NK1.1 expression is further defined into three groups based on CD27 and CD11b expression (*Hayakawa and Smyth, 2006*; *Kim et al., 2002b*). Adoptive transfer experiments have demonstrated that $CD27^+CD11b^-$ (CD27 single positive, SP) NK cells differentiate into $CD27^+CD11b^+$ (double positive, DP) cells. DP NK cells down-regulate CD27 to mature into $CD27^-CD11b^+$ (CD11b SP) cells, the terminally matured NK cell population which expresses KLRG1 (*Chiossone et al., 2009*; *Huntington et al., 2007*).

NK cell differentiation is driven by stage-specific transcription factors including Eomesodermin (Eomes) and T-bet which play critical roles in distinct maturation stages of NK cells (*Daussy et al., 2014*; *Gordon et al., 2012*; *Townsend et al., 2004*). Genetic knockout models demonstrated that Eomes is essential in driving maturation of CD27 SP to DP NK cells, while T-bet governs the terminal maturation of the DP to CD11b SP NK cells (*Gordon et al., 2012*; *Townsend et al., 2004*). Although the transcription factors involved in NK cell development at different maturation stages are established, the intracellular signaling pathways that regulate these transcription factors have not been well-defined.

mTOR is an evolutionarily conserved serine/threonine kinase that forms two functionally distinct complexes, mTOR complex 1 (mTORC1) and mTOR complex 2 (mTORC2) (*Saxton and Sabatini, 2017*; *Wullschleger et al., 2006*). Raptor and Rictor are the defining components of mTORC1 and mTORC2, respectively (*Hara et al., 2002*; *Jacinto et al., 2004*; *Kim et al., 2002a*; *Sarbassov et al., 2004*). These complexes regulate cell growth, proliferation, and metabolism and play an indispensable role in immune cells (*Powell et al., 2012*; *Saxton and Sabatini, 2017*; *Weichhart et al., 2015*; *Wullschleger et al., 2006*). The activation of mTORC1 involves the canonical PI3K-PDK1-Akt-TSC1/2-mTORC1 pathway (*Saxton and Sabatini, 2017*; *Yang and Guan, 2007*). Although less is known about the upstream activator of mTORC2, $PtdIns(3,4,5)P_3$, which is generated by PI3K (*Whitman et al., 1988*), has been shown to be critical (*Liu et al., 2015*). mTORC2 is known to phosphorylate Akt at Serine 473 and increase its kinase activity (*Guertin et al., 2006*). Whether this increased Akt kinase activity mediated by mTORC2 is critical for mTORC1 activation in physiological content remains unknown.

Signaling through IL-15 receptors is obligatory for NK cell development (*DiSanto et al., 1995*; *Kennedy et al., 2000*; *Suzuki et al., 1997*), and activation of PI3K is one component of IL-15 receptors signaling (*Gu et al., 2000*; *Zhu et al., 1994*). These imply that both mTORC1 and mTORC2 are essential for NK cell development. Indeed, $Ncr1^{iCre}$-mediated NK-cell-specific ablation of mTOR results in impaired development and effector functions (*Marçais et al., 2014*). While this study reveals the requirement of mTOR itself, the independent contributions of mTORC1 or mTORC2 in NK cell development remain undefined. The primary limitation of interpreting mTOR deficiency is the inability to attribute the functional and developmental outcomes to precise signaling pathways and transcription factors that are uniquely regulated by either mTORC1 or mTORC2. Previous studies have demonstrated that PDK1, an upstream activator of mTORC1, regulates early NK cell development by inducing Nfil3 which drives the transcription of *Eomes* (*Yang et al., 2015*). FoxO1, a downstream effector of mTORC2, negatively regulates the terminal maturation of NK cells through direct inhibition of the transcription of *Tbx21* (the gene encoding T-bet) (*Deng et al., 2015*). Based on these, we hypothesized that mTORC1 and mTORC2 regulate NK cell development through differentially driving the expression of distinct T-box transcription factors.

To test this hypothesis, we conditionally deleted Raptor (*Rptor*) or Rictor (*Rictor*) to eliminate the formation of mTORC1 or mTORC2, respectively, in NK cells by using $Ncr1^{iCre}$ transgenic mice (*Narni-Mancinelli et al., 2011*). We find both mTORC1 and mTORC2 are essential for NK cell development. The ablation of mTORC1 disrupts NK cell homeostasis and blocks the transition of the CD27 SP to DP stage. RNAseq analyses reveal significant alteration of the gene expression profile in Raptor-deficient NK cells with impaired expression of transcription factors governing early NK cell development. Loss of mTORC2 blocks the transition of the DP to the terminally mature CD11b SP NK cells. This defect is associated with impaired induction of T-bet through a mechanism involving the $mTORC2-Akt^{S473}-FoxO1$ signaling axis. These findings reveal distinct roles of mTORC1 and mTORC2 in NK cell development and define them as the upstream regulators of T-box transcription factors during NK cell development.

## Results

### mTORC1 is critical for homeostasis and differentiation of NK cells

To define the role of mTORC1 in NK cell homeostasis, we generated NK-cell-specific conditional knockout (cKO) mice by breeding $Rptor^{fl/fl}$ mice (*loxp* sites targeting exon 6) with $Ncr1^{iCre}$ knockin mice. Expression of Cre driven by $Ncr1$ promoter resulted in the deletion of $Rptor$ and functional loss of mTORC1 during the immature NK cell stage. Loss of Raptor protein in NK cells was verified by western blot (*Figure 1A*). Phenotypic analyses revealed that the frequency of NK cells was increased in the BM of $Rptor$ cKO compare to WT mice, while percentages of NK cells in the periphery were significantly reduced (*Figure 1B*). The lymphocytes counts were comparable between WT and $Rptor$ cKO mice in both BM and spleen (*Figure 1—figure supplement 1A*).

To explain the reduced NK cell number in the periphery of $Rptor$ cKO mice, we investigated cell proliferation, migration, and viability. The percentage of proliferating NK cells was significantly reduced in $Rptor$ cKO mice at steady-state, as evidenced by Ki-67 staining (*Figure 1C*). Increased cell number in the BM suggested a potential impairment in the trafficking of NK cells. To test this, WT and $Rptor$ cKO mice were intravenously injected with an anti-CD45.2 antibody, sacrificed after 2 min, and their BM cells were analyzed. This allowed us to quantify the number of NK cells in the sinusoidal versus parenchymal regions of the BM, an indicator of NK cell trafficking under steady state (*Leong et al., 2015*). The frequency and number of CD45.2$^+$ NK cells were significantly reduced in $Rptor$ cKO mice, indicating impairment in the trafficking of NK cells (*Figure 1D*). There were no differences in cell viability between WT and $Rptor$ cKO NK cells, as demonstrated by Annexin V and Propidium iodide staining (*Figure 1—figure supplement 1B*). These data showed that disruption of mTORC1 impairs homeostatic NK cell proliferation and migration, but not viability.

Next, we investigated the role of mTORC1 in NK cell differentiation. Expression of CD122, NK1.1, and DX5 indicated a reduction of the mNK population in the spleen, while the iNK population was similar between WT and $Rptor$ cKO mice (*Figure 1—figure supplement 1C*), which matches with the onset of $Ncr1$ expression occurring at the late stage of iNK. No significant changes were observed among NKPs, iNKs, and mNKs in the BM (*Figure 1—figure supplement 1C*). We then focused our analyses on NK cell maturation using cell surface markers CD27 and CD11b. Raptor deficiency resulted in a significant block in the transition from the CD27 SP to DP stage in both the BM and periphery (*Figure 1E*, *Figure 1—figure supplement 1D and E*). Consistent with this, the frequency of KLRG1-expressing NK cells was also significantly reduced in all organs tested (*Figure 1F*). Analyses of other activating and inhibitory cell surface receptors indicated further developmental defects including reduced expression of Ly49D, Ly49G2, and Ly49H (*Figure 1—figure supplement 1F*). Collectively, we conclude that mTORC1 is required for maintaining NK cell homeostasis through proliferation, migration, and differentiation.

### mTORC2 is required for terminal maturation of NK cells

To elucidate the role of mTORC2 in NK cell development, we crossed $Rictor^{fl/fl}$ mice (*loxp* sites targeting exon 11) with $Ncr1^{iCre}$ mice to generate conditional knockout of $Rictor$ in NK cells. The loss of Rictor protein was verified by western blot (*Figure 2A*). Unlike $Rptor$ cKO mice, loss of mTORC2 did not change frequency or number of NK cells in the BM (*Figure 2B*); however, the percentages of NK cells were drastically reduced in the periphery with the exception of inguinal lymph nodes (*Figure 2B*). The absolute numbers of lymphocytes were comparable between WT and $Rictor$ cKO mice in both BM and spleen (*Figure 2—figure supplement 1A*). The reasons for the reduction in NK cell numbers in $Rictor$ cKO mice could be due to altered proliferation, migration or viability. Ki-67 staining indicated the frequency of proliferating NK cells at steady-state was significantly reduced in $Rictor$ cKO compare to WT mice (*Figure 2C*). Although the total numbers of NK cells in the BM were similar (*Figure 2B*), CD45.2 staining revealed a significantly higher number of NK cells in the parenchymal region in $Rictor$ cKO compare to WT mice, implying a potential defect in their migration (*Figure 2D*). Rictor deficiency did not affect the viability of NK cells (*Figure 2—figure supplement 1B*). Together, we conclude that loss of mTORC2 results in defective NK cell proliferation and migration that lead to a reduction of their numbers in the periphery.

The reduced total NK cell number in the spleen of $Rictor$ cKO mice was exclusively associated with a reduction in the mNK population (*Figure 2—figure supplement 1C*). Similarly, there are no

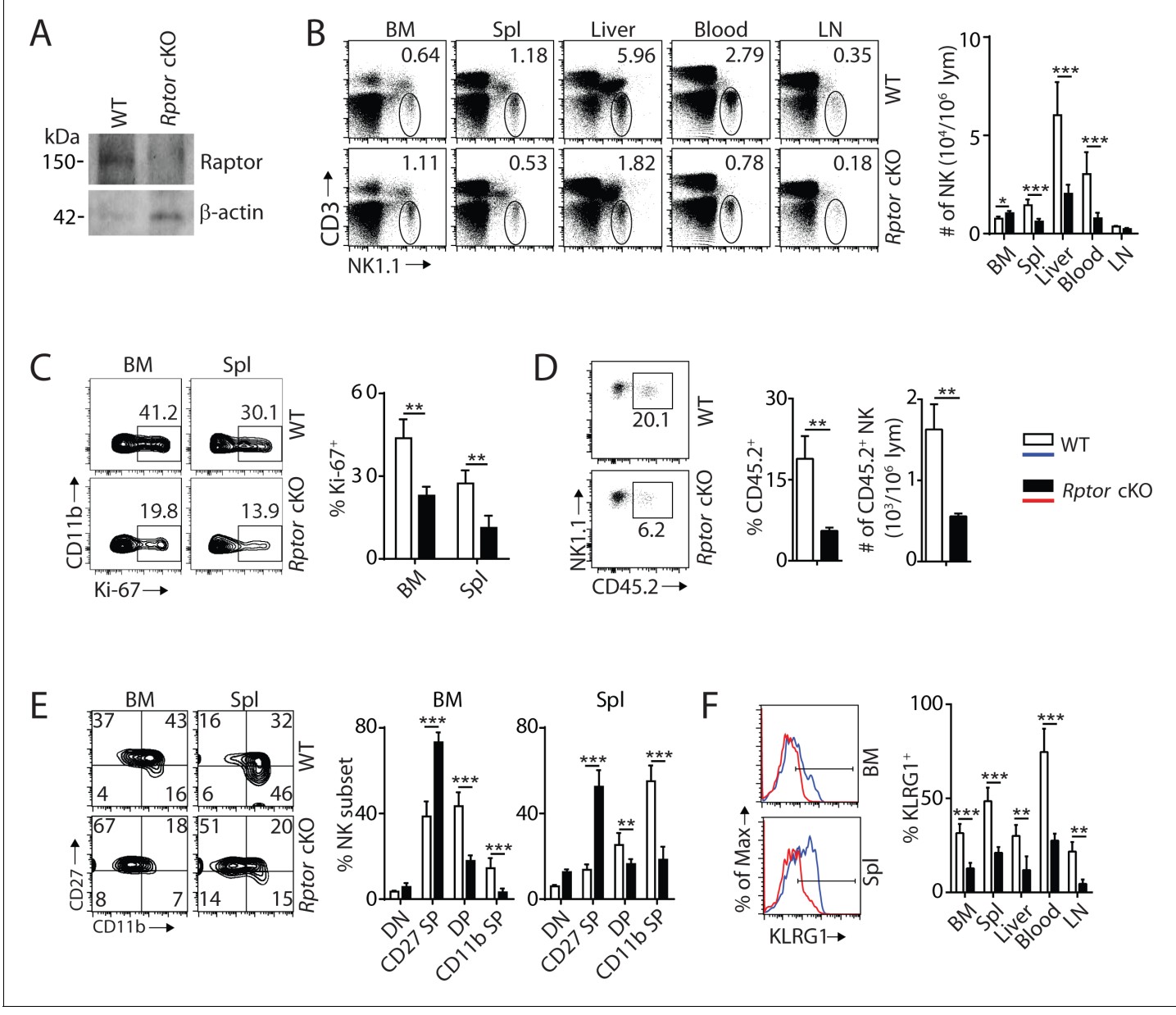

**Figure 1.** mTORC1 is essential for NK cell homeostasis and maturation. (**A**) Raptor expression in freshly-isolated NK cells from WT and *Rptor* cKO mice was evaluated via western blot. (**B**) CD3, NK1.1 staining of cells from various organs (left) and quantification of NK cells in each organ of WT and *Rptor* cKO mice (right). n = 4–8 pooled from two to four independent experiments. (**C**) Ki-67 staining was used to assess steady-state proliferation of NK cells gated on CD11b⁻ population (left), and percentage of Ki-67⁺ cells (right). n = 3 pooled from three independent experiments. (**D**) Percentage of NK cells that are in the sinusoidal compartment of BM was demonstrated by CD45.2 staining (left) and quantified as both percentage and number of CD45.2⁺ NK cells per million lymphocytes (right). n = 3 pooled from three independent experiments. (**E**) CD27 and the CD11b expression on gated NK cells from BM and spleen of WT and *Rptor* cKO mice were assessed by flow cytometry (left), and percentages of each NK subsets were quantified (right). n = 7 pooled from three independent experiments. (**F**) The KLRG1 expression on gated NK cells from BM and spleen of WT and *Rptor* cKO mice (left) and percentage of KLRG1⁺ cells within NK populations from different organs (right). n = 4 pooled from two independent experiments. All bar graphs present the mean ± SD. Statistical significance was calculated using two-way ANOVA (**B, C, E, F**) or unpaired Student t-test (**D**). *p<0.05; **p<0.01; ***p<0.001.

DOI: https://doi.org/10.7554/eLife.35619.002

The following figure supplement is available for figure 1:

**Figure supplement 1.** mTORC1 is required for NK cell maturation.

DOI: https://doi.org/10.7554/eLife.35619.003

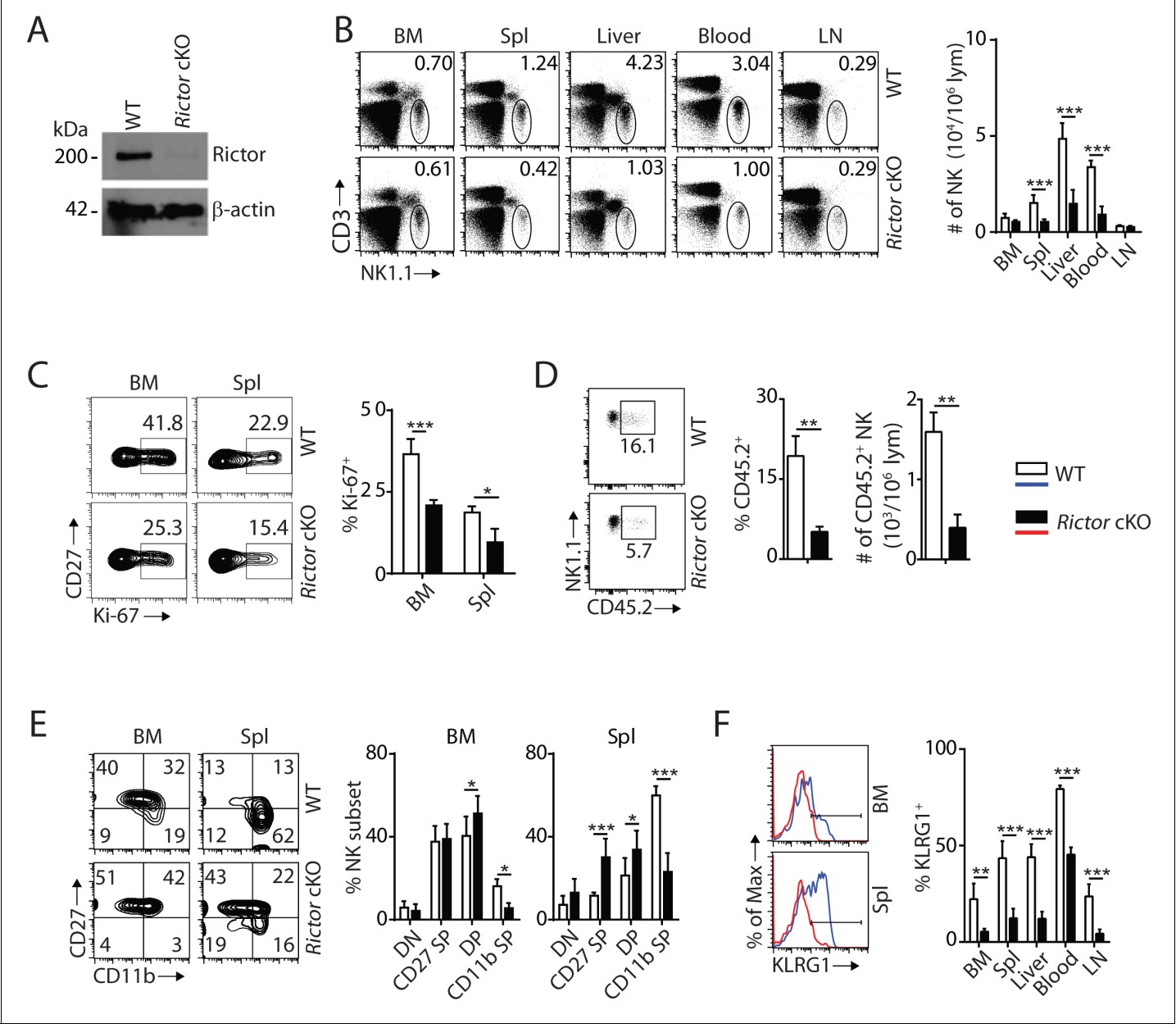

**Figure 2.** mTORC2 is pivotal for NK cell homeostasis and terminal maturation. (A) Rictor expression in IL-2-cultured NK cells isolated from WT and *Rictor* cKO mice was evaluated via western blot. A representative of three independent experiments. (B) CD3, NK1.1 staining of cells from various organs (left) and quantification of NK cells in each organ of WT and *Rictor* cKO mice (right). n = 4–7 pooled from two to four independent experiments. (C) Ki-67 staining was used to assess steady-state proliferation of NK cells gated on CD27$^+$ population (left), and percentage of Ki-67$^+$ cells (right). n = 3 pooled from three independent experiments. (D) Percentage of NK cells that are in the sinusoidal compartment of BM was demonstrated by CD45.2 staining (left) and quantified as both percentage and number of CD45.2$^+$ NK cells per million lymphocytes (right). n = 3 pooled from three independent experiments. (E) CD27 and the CD11b expression on gated NK cells from BM and spleen of WT and *Rictor* cKO mice were assessed by flow cytometry (left), and percentages of each NK subsets were quantified (right). n = 6 pooled from four independent experiments. (F) The KLRG1 expression on gated NK cells from BM and spleen of WT and *Rictor* cKO mice (left) and percentage of KLRG1$^+$ cells within NK populations from different organs (right). n = 3–4 pooled from two or three independent experiments. All bar graphs present the mean ± SD. Statistical significance was calculated using two-way ANOVA (B, C, E, F) or unpaired Student t-test (D). *p<0.05; **p<0.01; ***p<0.001.

DOI: https://doi.org/10.7554/eLife.35619.004

The following figure supplement is available for figure 2:

**Figure supplement 1.** Terminal NK cell maturation was impaired in *Rictor* cKO mice.

DOI: https://doi.org/10.7554/eLife.35619.005

significant changes among NKPs, iNKs, and mNKs in the BM (*Figure 2—figure supplement 1C*). Distinct from *Rptor* cKO mice where NK cell maturation was blocked at the CD27 SP stage, loss of Rictor impaired NK cell maturation from the DP stage to CD11b SP stage in both the BM and peripheral organs (*Figure 2E*, *Figure 2—figure supplement 1D and E*). This defect in terminal maturation was further validated by a significant reduction in KLRG1-expressing NK cells in all organs tested (*Figure 2F*). Loss of Rictor also resulted in an altered expression profile of cell surface receptors in NK cells. Expression of activating or inhibitory Ly49 receptors was reduced, while NK1.1 and DX5 expression were increased on *Rictor* cKO NK cells (*Figure 2—figure supplement 1F*). We conclude that mTORC2 is required for the terminal maturation of NK cells, and therefore regulates NK cell development at a different stage than that of mTORC1.

## mTORC1 and mTORC2 differentially regulate the expression of T-box transcription factors

T-box transcription factors Eomes and T-bet are critical for NK cell maturation (*Daussy et al., 2014*; *Gordon et al., 2012*; *Townsend et al., 2004*). Accumulation of CD27 SP NK cells in *Rptor* cKO mice (*Figure 1E*, *Figure 1—figure supplement 1D and E*) was similar to earlier findings in *Eomes^{fl/fl} Vav-^{Cre}* mice or Eomes-negative NK cells that naturally occur in the liver of WT mice (*Gordon et al., 2012*). This prompted us to evaluate the expression of Eomes in Raptor-deficient NK cells. Intracellular staining revealed a significant reduction in the protein levels of Eomes in CD27 SP, DP and CD11b SP NK cells from *Rptor* cKO mice (*Figure 3A*). Compare to Eomes, the expression of T-bet was minimal in NK cells within the BM (*Figure 3B*) that was consistent with earlier reports (*Daussy et al., 2014*). Loss of Raptor resulted in a moderate reduction in T-bet protein level in NK cells from the spleen (*Figure 3B*). On the other hand, the normal developmental progression from the CD27 SP to DP stage (*Figure 2E*, *Figure 2—figure supplement 1D and E*) correlated with unaltered expression of Eomes in Rictor-deficient NK cells (*Figure 3C*). The defect in the terminal maturation of NK cells in *Rictor* cKO mice mimicked the maturation defects seen in *Tbx21* KO mice (*Gordon et al., 2012*; *Townsend et al., 2004*). We found the expression of T-bet indeed is significantly reduced in NK cells from the spleen of *Rictor* cKO mice (*Figure 3D*).

## mTORC1 and mTORC2 differentially regulate the expression of CD122 and STAT5 activation

Eomes binds to the promoter of *Il2rb* (gene encoding IL-15/IL-2 receptor β chain, CD122) and activates its expression (*Intlekofer et al., 2005*). Given that Eomes expression is down-regulated (*Figure 3A*), we asked whether or not CD122 expression is impaired in *Rptor* cKO mice. Flow analyses revealed that the expression of CD122 was significantly reduced on per cell basis in *Rptor* cKO compared to WT mice in all three subsets of NK cells from BM and spleen (*Figure 4A*). The expression of another subunit of the IL-15 receptor complex, the common γ chain (CD132), was not altered in *Rptor* cKO NK cells (*Figure 4—figure supplement 1A*). The reduction of CD122 expression in *Rptor* cKO NK cells also led to reduced STAT5 phosphorylation following ex vivo IL-15 stimulation (*Figure 4B*). These results indicate that besides mTORC1-mediated signaling, other passways downstream of IL-15 receptors were sub-optimal in *Rptor* cKO NK cells, which may also have potentially contributed to the developmental defects.

Previous reports have shown that mTORC1 activity is reduced during the transition from the DP to CD11b SP stage of WT mice (*Marçais et al., 2014*). We validated this observation by comparing phosphorylation of rpS6^{S240/244} and cell size between these two NK subsets from WT mice (*Figure 4—figure supplement 1B and C*). Concurrently, we found that the protein levels of Eomes were also reduced during this transition (*Figure 4—figure supplement 1D*). We also observed a similar reduction in CD122 expression in the CD11b SP compare to the DP NK cells from WT mice (*Figure 4—figure supplement 1E*). These phenotypical changes during the transition from the DP to CD11b SP stage of WT mice mimic Raptor-deficient NK cells. This implies a regulation loop among mTORC1 → Eomes → CD122 → mTORC1 that regulate the development of normal NK cells.

Similar to Eomes, T-bet has also been shown to regulate CD122 expression in T cells (*Intlekofer et al., 2005*). Therefore, we evaluated the CD122 expression and IL-15 receptor signaling in *Rictor* cKO NK cells. Unlike *Rptor* cKO NK cells that had significant reduction in CD122 expression (*Figure 4A*), *Rictor* cKO NK cells exhibited only a moderate reduction (*Figure 4C*), which was

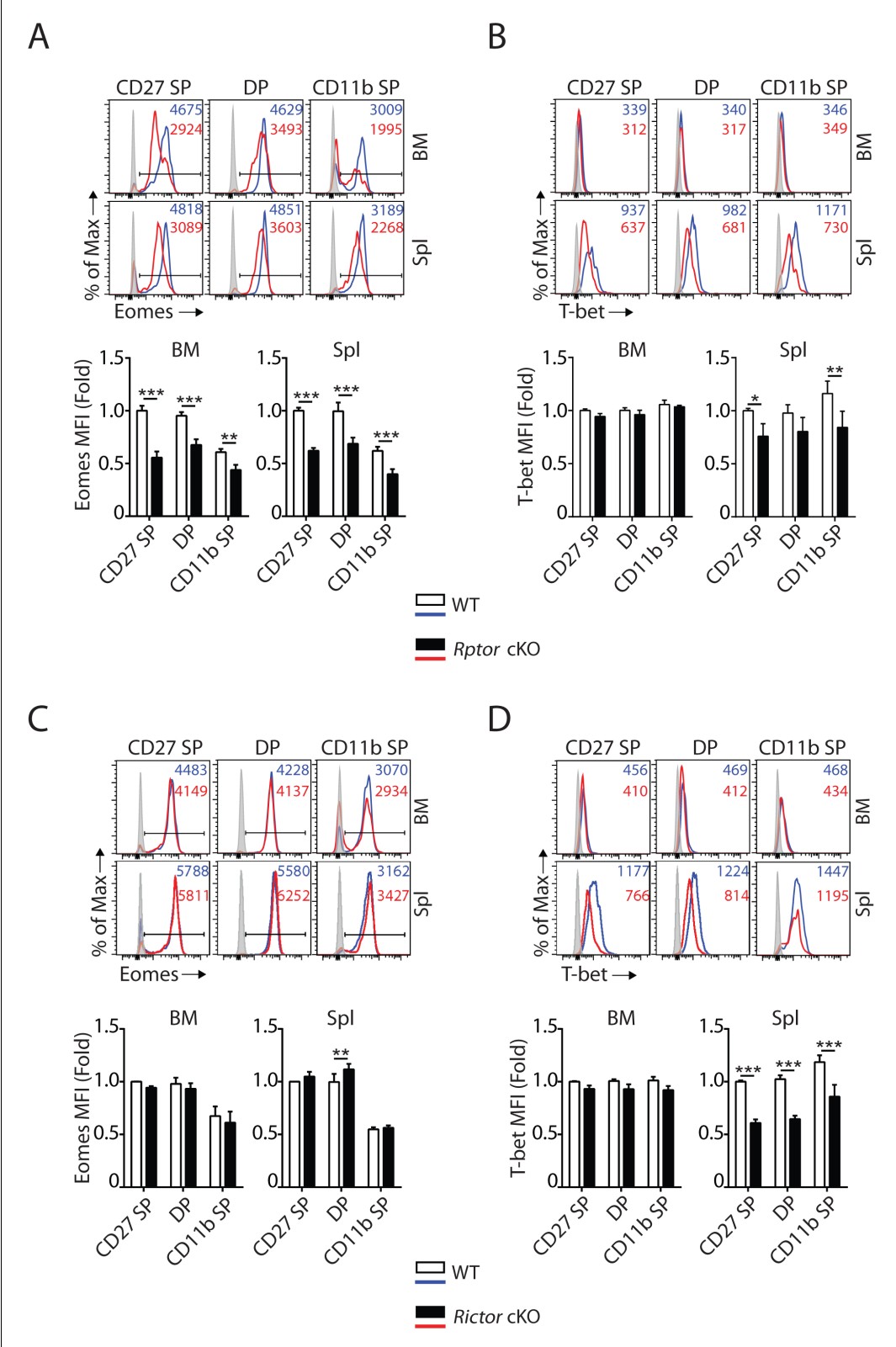

**Figure 3.** mTORC1 and mTORC2 differentially regulate the expression of Eomes and T-bet. (A–D) Histogram of Eomes (A, C) and T-bet (B, D) expression on each NK cell population gated by CD27 and CD11b of *Rptor* (A, B) or *Rictor* (C, D) cKO mice and their corresponding WT control. The histogram in grey presents the unstained control (top). Mean fluorescent intensity (MFI) is shown as fold change normalized to WT CD27 SP population
*Figure 3 continued on next page*

*Figure 3 continued*

(bottom). n = 4 pooled from three independent experiments. All bar graphs present the mean ± SD. Statistical significance was calculated using two-way ANOVA. *p<0.05; **p<0.01; ***p<0.001.

DOI: https://doi.org/10.7554/eLife.35619.006

consistent with the notion that Eomes, but not T-bet, plays a critical role in maintaining CD122 expression in NK cells (*Intlekofer et al., 2005*; *Townsend et al., 2004*). The expression of CD132 is not altered in Rictor-deficient NK cells (*Figure 4—figure supplement 1F*). In addition, IL-15-mediated STAT5 phosphorylation was intact in *Rictor* cKO NK cells (*Figure 4D*).

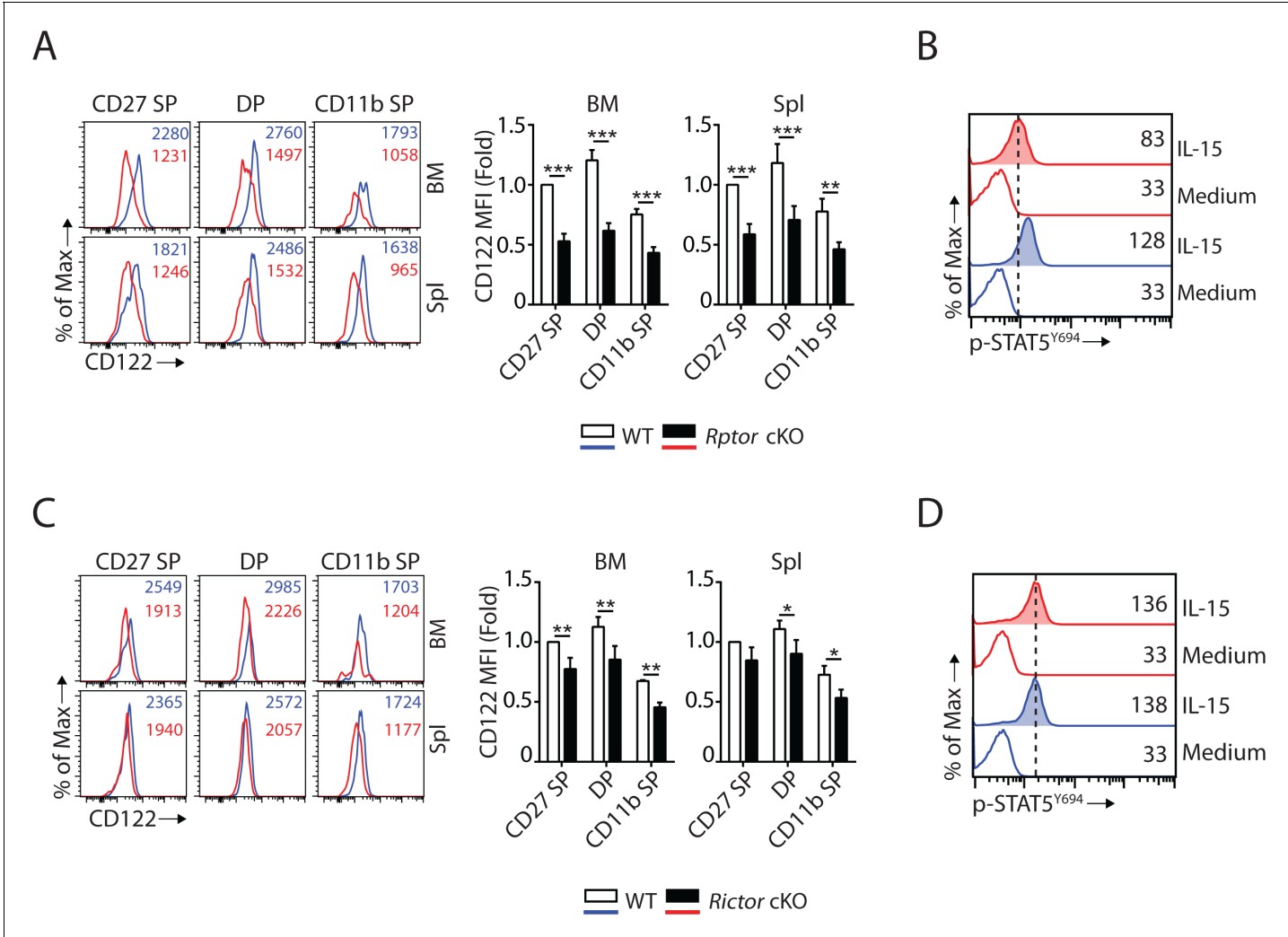

**Figure 4.** mTORC1 and mTORC2 differentially regulate the expression of CD122 and STAT5 activation. (**A**) Histogram of CD122 expression on each NK population gated by CD27 and CD11b of WT and *Rptor* cKO mice (left). MFI of CD122 was normalized to WT CD27 SP population (right). n = 3 pooled from three independent experiments. (**B**) Splenocytes from WT or *Rptor* cKO mice were stimulated with either medium or 100 ng/mL IL-15 for 1 hr. Phosphorylation of STAT5$^{Y694}$ was detected by phosphor-flow and shown as the representative histogram of three independent experiments. (**C, D**) CD122 expression (**C**) and IL-15-mediated STAT5 activation (**D**) in WT or Rictor-deficient NK cells. Same experimental procedures as A and B. All bar graphs present the mean ± SD. Statistical significance was calculated using two-way ANOVA. *p<0.05; **p<0.01; ***p<0.001.

DOI: https://doi.org/10.7554/eLife.35619.007

The following figure supplement is available for figure 4:

**Figure supplement 1.** mTORC1 activity, Eomes and CD122 expression were concurrently reduced during the transition from DP to CD11b SP stage.

DOI: https://doi.org/10.7554/eLife.35619.008

## Gene network analyses of NK cells from *Rptor* or *Rictor* cKO mice

To further explore the mechanism on how mTORC1 and mTORC2 differentially regulate the development of NK cells, we performed RNAseq analyses using FACS-sorted BM NK cells from *Rptor* cKO, *Rictor* cKO and their respective WT mice (n = 3 per group). To reduce the developmental bias, we sorted CD11b⁻ BM NK cells from littermate WT and *Rptor* cKO mice or CD27⁺ BM NK cells from littermate WT and *Rictor* cKO mice. Compared to their corresponding WT, 2963 genes in *Rptor* cKO NK cells and 829 genes in *Rictor* cKO NK cells were differentially expressed (DE; FDR < 0.05). Of which 406 DE genes were overlapped between *Rptor* and *Rictor* cKO NK cells, indicating a more profound transcriptomic alteration in *Rptor* cKO NK cells compared to *Rictor* cKO NK cells (*Figure 5A*). After normalizing the level of each transcript in the *Rptor* or *Rictor* cKO NK cells to its corresponding WT, we plotted all the genes using the volcano plots, demonstrating the overall change in the transcriptomic profile (*Figure 5B and C*). The orange/red dots represent genes that are significantly increased, while the aqua/dark blue dots represent genes that are significantly decreased in Raptor- and Rictor-deficient NK cells compared to the corresponding WT counterparts. Several key transcripts are highlighted in *Figure 5B and C*.

Ingenuity Pathway Analysis (IPA) revealed distinct gene ontology enrichment in *Rptor* and *Rictor* cKO NK cells. As downstream targets of mTORC1, the top three significantly enriched signaling pathways in *Rptor* cKO NK cells are eIF2, eIF4/P70S6K, and mTOR signaling (p=5.53 × $10^{-23}$, 2.15 × $10^{-14}$, and 4 × $10^{-14}$, respectively), consistent with mTORC1 being the bona fide regulator of protein synthesis (*Thoreen et al., 2012*). Unexpectedly, the Z-scores of those three pathways are positive (Z-score = 3.833, 0.894, and 0.73, respectively), which indicates higher translational activity in *Rptor* cKO NK cells. After a thorough examination of the molecular signature of those three enriched pathways, we found that Raptor deficiency results in increased transcription of genes encoding proteins that comprise the translation machinery such as eIFs, eEFs and ribosome proteins (*Figure 5D*, *Figure 5—source data 1*). This not only explains the positive Z-scores of those three pathways but also reveals that compensatory pathways are initiated in *Rptor* cKO NK cells to overcome the impaired protein translation.

IPA analyses also demonstrated impaired oxidative phosphorylation pathway in *Rptor* cKO NK cells, indicating defects in mitochondrial functions (*Figure 5—figure supplement 1A*). This coincides with more than 10-fold induction of *Ppargc1a* in *Rptor* cKO NK cells (*Figure 5B*), which is known to be induced under mitochondrial stress conditions (*Finck and Kelly, 2006*). Besides oxidative phosphorylation, integrin signaling is also defective in *Rptor* cKO NK cells (*Figure 5—figure supplement 1B*). As for *Rictor* cKO NK cells, the PTEN signaling pathway is impaired (p=1.73 × $10^{-5}$, Z-score = −1.069) as demonstrated by increased expression of receptors (*Insr, Igf1r*) involving growth factors signaling or proteins (*Pik3cd, Pik3r5*) comprising PI(3)K and decreased expression of phosphatase (*Inpp5b*) that dampen the inositol phosphates signaling (*Figure 5E*). This indicates that there is a potential positive regulation of PTEN by mTORC2 to balance the PI(3,4,5)P₃-mediated activation of mTORC2 (*Liu et al., 2015*).

## Altered transcriptome relates to impaired NK cell development

Using RNA sequencing data, we examined the expression levels of key transcription factors governing different developmental stages of NK cells. The expression of transcription factors such as Nfil3, Id2, and Eomes that are critical for NK cell commitment and early development are reduced in *Rptor* but not in *Rictor* cKO NK cells (*Table 1*, *Figure 5—figure supplement 1C*). The expression of T-bet and Zeb2 which promote terminal mature NK cell development were significantly reduced in *Rictor* cKO mice (*Table 1*, *Figure 5—figure supplement 1C*). This expression pattern correlates with early NK cell development impairment in *Rptor* cKO mice versus terminal maturation defect seen in *Rictor* cKO mice (*Figures 1E* and *2E*).

Although Nfil3 has been proposed to induce Id2 and Eomes (*Male et al., 2014*), *Ncr1^iCre*-mediated deletion of *Nfil3* does not alter the development of NK cells (*Firth et al., 2013*). Therefore, we next focused our analyses on Id2 and Eomes since deficiency of either of them renders similar developmental defects as in the *Rptor* cKO mice (*Delconte et al., 2016*; *Gordon et al., 2012*). Delconte et al. have shown that Id2 is critical to suppress the expression of E-protein target genes during NK cell development. However, we did not find induction of those E-protein target genes in *Rptor* cKO NK cells (*Supplementary file 1*). On the contrary, compared with the WT controls, we found a

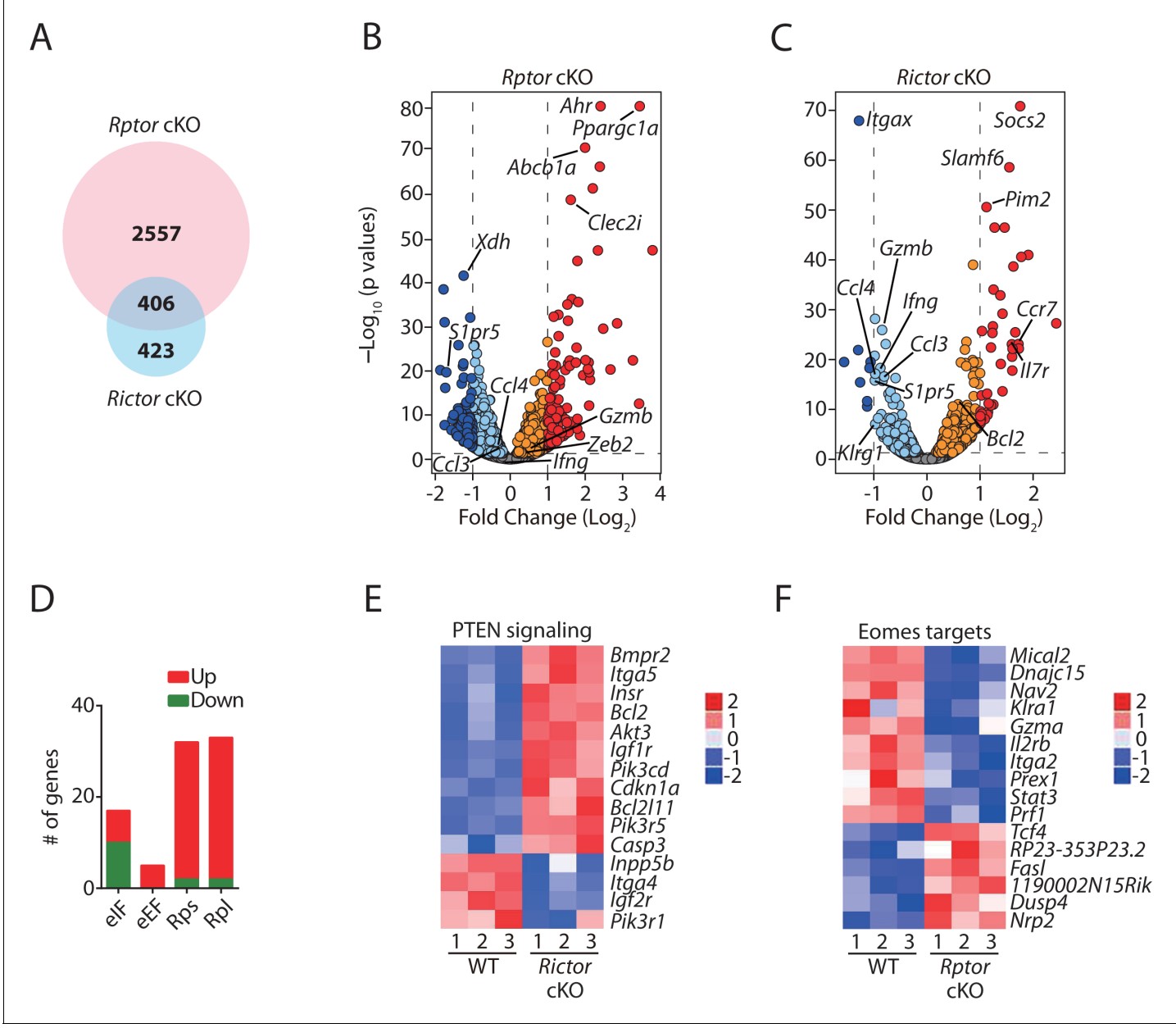

**Figure 5.** Transcriptome analyses of Raptor- or Rictor-deficient NK cells. RNAseq was conducted using CD11b⁻ BM NK cells from littermate WT and *Rptor* cKO mice or CD27⁺ BM NK cells from littermate WT and *Rictor* cKO mice. (n = 3 per group). (**A**) Venn diagram demonstrating the number of genes that are differentially expressed (FDR < 0.05) in Raptor- or Rictor-deficient NK cells compared to their corresponding littermate WT control cells and the overlapping between those two genes list. (**B, C**) Volcano plot demonstrating the overall transcriptome alterations in Raptor- (**B**) or Rictor-deficient (**C**) NK cells compared to their corresponding littermate WT control cells. The orange/red dots represent genes that are significantly increased, while the aqua/dark blue dots represent genes that are significantly decreased in Raptor- and Rictor-deficient NK cells compared to the corresponding WT counterparts. We plotted all the genes with –Log₁₀(p values) greater than 80 at the y-axis equal to 80. (**D**) A number of significantly up-regulated or down-regulated genes encoding proteins belonging to eIFs, eEFs, ribosome small (Rps) or large (Rpl) subunits family in *Rptor* cKO NK cells are quantified and presented in the bar graph. (**E, F**) Enrichment of PTEN signaling in *Rictor* cKO NK cells (**E**) and Eomes target genes in *Rptor* cKO NK cells (**F**) were demonstrated via the heatmap.

DOI: https://doi.org/10.7554/eLife.35619.009

The following source data and figure supplement are available for figure 5:

**Source data 1.** Differentially expressed genes (FDR < 0.05) that belong to eIFs, eEFs and ribosome protein families in *Rptor* cKO NK cells was listed in the table.

DOI: https://doi.org/10.7554/eLife.35619.011

**Figure supplement 1.** Altered oxidative phosphorylation and integrin signaling in Raptor-deficient NK cells indicated by RNA sequencing.

*Figure 5 continued on next page*

*Figure 5 continued*

DOI: https://doi.org/10.7554/eLife.35619.010

significant enrichment of Eomes target genes in *Rptor* cKO NK cells (p=6.11 $\times$ 10$^{-3}$, Z-score = −0.988) (**Figure 5F**). This indicates that the deficiency of Eomes in *Rptor* cKO NK cells is more likely to be responsible for the developmental defects.

In addition, we identified that both Raptor and Rictor deficiency results in reduced transcripts level of *S1pr5* (**Figure 5B and C**). This further implies the migration defects of both Raptor- and Rictor-deficient NK cells. Moreover, an increase in the transcript level of *Socs2* in *Rictor* cKO NK cells (**Figure 5C**) may contribute to their impaired proliferation and reduced cellularity (**Kim et al., 2017**). Related to NK cell effector functions, the RNA sequencing data revealed that Rictor, but not Raptor, deficiency results in reduced transcripts level of *Ifng* and *Gzmb* (**Figure 5B and C**). In fact, the mRNA level of Granzyme B is significantly increased in *Rptor* cKO NK cells (**Figure 5B**). Loss of Raptor or Rictor impaired the expression of Ccl3 and Ccl4 (**Figure 5B and C**).

## Hyperactive FoxO1 potentially results in impaired T-bet expression in *Rictor* cKO NK cells

Consistent with reduced T-bet expression in mRNA (**Table 1**) and protein level (**Figure 3D**), we found a significant enrichment of T-bet target genes (p=4.59 $\times$ 10$^{-14}$, Z-score = −1.429) in *Rictor* cKO NK cells (**Figure 6A**). The well-established T-bet-induced genes (*Klrg1*, *Ifng*, *Gzmb*, *S1pr5*, *Zeb2*) (**Simonetta et al., 2016**) were reduced in *Rictor* cKO NK cells (**Figures 6A** and **5C**). Next, we seek to uncover the mechanism through which mTORC2 regulates the expression of T-bet. Earlier work showed that mTORC2 phosphorylates Serine$^{473}$ on Akt, which is critical for Akt to phosphorylate FoxO transcription factors (**Brunet et al., 1999**; **Guertin et al., 2006**). After Akt-mediated

**Table 1.** mRNA level of key transcription factors governing NK cell development in *Rptor* or *Rictor* cKO NK cells.

| | | Raptor | | | | | | |
|---|---|---|---|---|---|---|---|---|
| List | Gene | WT mean | WT se | cKO mean | cKO se | log2FC | Adjusted p value | Significant at FDR 0.05 |
| key TF | Id2 | 1 | 0.0116 | 0.7037 | 0.0393 | −0.492 | 1.01E-06 | Y |
| | Nfil3 | 1 | 0.024 | 0.7521 | 0.1014 | −0.379 | 0.0369946 | Y |
| | Eomes | 1 | 0.0246 | 0.758 | 0.0111 | −0.392 | 1.77E-06 | Y |
| | Gata3 | 1 | 0.0195 | 0.9258 | 0.0542 | −0.105 | 0.5441197 | N |
| | Ets1 | 1 | 0.0247 | 0.6935 | 0.0272 | −0.515 | 1.37E-08 | Y |
| | Tox | 1 | 0.0549 | 1.0158 | 0.0409 | 0.02 | 0.9239868 | N |
| | Tbx21 | 1 | 0.0295 | 0.7811 | 0.0306 | −0.346 | 0.0009024 | Y |
| | Zeb2 | 1 | 0.0826 | 1.5004 | 0.1377 | 0.546 | 0.0005179 | Y |
| | Prdm1 | 1 | 0.0925 | 1.1062 | 0.2453 | 0.102 | 0.8261435 | N |
| | | Rictor | | | | | | |
| List | gene | WT mean | WT se | cKO mean | cKO se | log2FC | Adjusted p value | Significant at FDR 0.05 |
| key TF | Id2 | 1 | 0.1133 | 1.0699 | 0.0271 | 0.088 | 0.8182466 | N |
| | Nfil3 | 1 | 0.0354 | 0.9495 | 0.0635 | −0.069 | 0.8687849 | N |
| | Eomes | 1 | 0.0408 | 1.0366 | 0.0362 | 0.049 | 0.8800234 | N |
| | Gata3 | 1 | 0.0118 | 1.0105 | 0.0283 | 0.014 | 0.9839612 | N |
| | Ets1 | 1 | 0.0516 | 0.8081 | 0.0529 | −0.287 | 0.0117066 | Y |
| | Tox | 1 | 0.0457 | 0.7144 | 0.0379 | −0.442 | 0.0001058 | Y |
| | Tbx21 | 1 | 0.0562 | 0.7603 | 0.073 | −0.357 | 0.0065806 | Y |
| | Zeb2 | 1 | 0.0084 | 0.7995 | 0.0323 | −0.298 | 0.0137732 | Y |
| | Prdm1 | 1 | 0.1397 | 1.0292 | 0.1234 | 0.026 | 0.9839612 | N |

DOI: https://doi.org/10.7554/eLife.35619.012

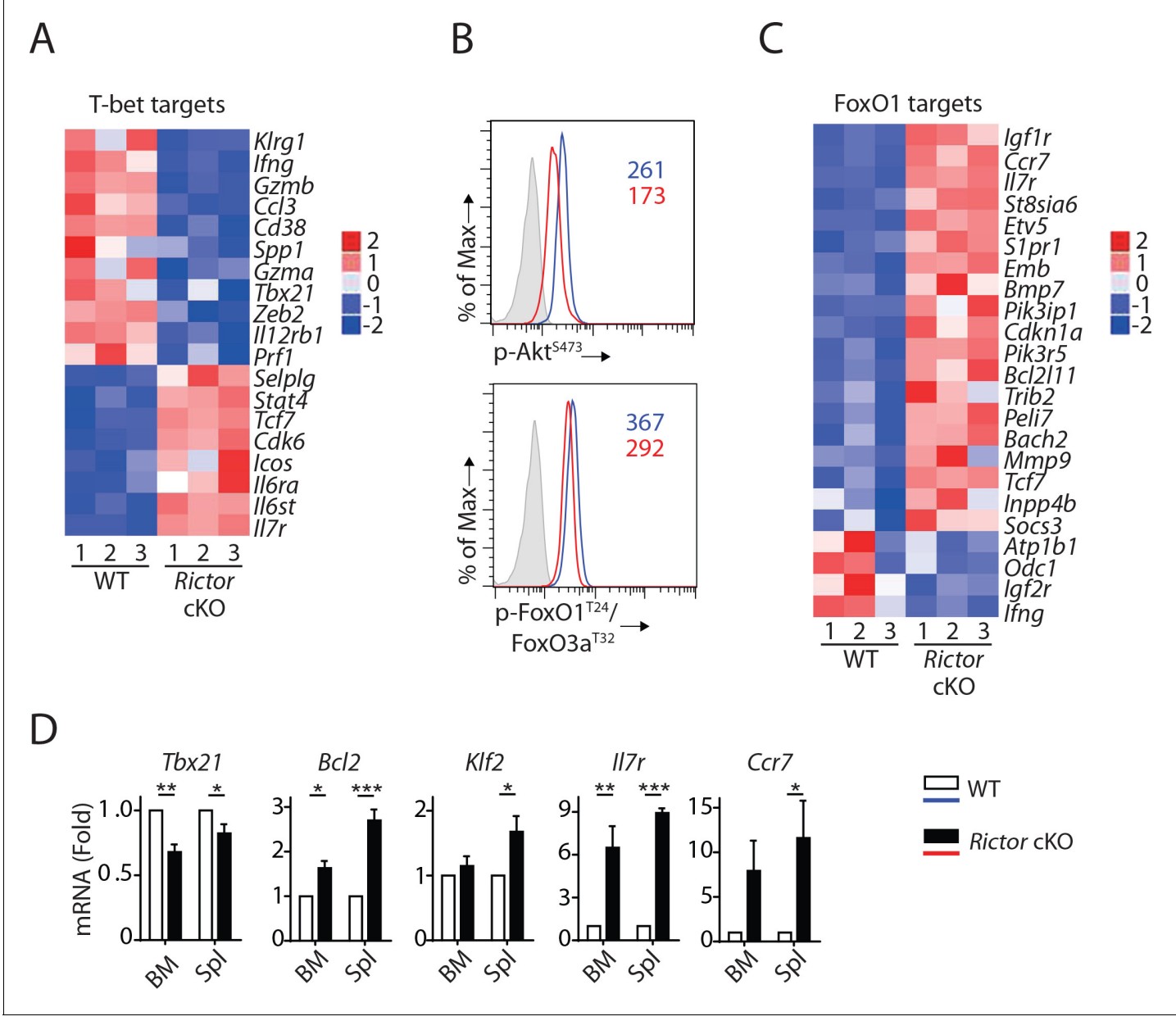

**Figure 6.** mTORC2 is required for T-bet expression through regulation of FoxO1 during NK cell development. (**A**) Enrichment of T-bet target genes in *Rictor* cKO NK cells as shown via the heatmap. (**B**) Histogram demonstrating phosphorylation of Akt$^{S473}$ (top) and FoxO1$^{T24}$/FoxO3a$^{T32}$ (bottom) in NK cells from BM of WT or *Rictor* cKO mice. The histogram in grey presents the isotype control. A representative of two or three independent experiments. (**C**) Enrichment of FoxO1 target genes in *Rictor* cKO NK cells as shown via the heatmap. (**D**) The mRNA level of *Tbx21*, *Bcl2*, *Klf2*, *IL7r*, and *Ccr7* were evaluated by RT-qPCR with sorted fresh CD27$^+$ NK cells from BM and spleen of WT or *Rictor* cKO mice. The data were shown as fold change normalized to WT. n = 3–5 pooled from three to five independent experiments. All bar graphs present the mean ± SD expect (**D**) which is shown as the mean ± SEM. Statistical significance was calculated using two-way ANOVA. *p<0.05; **p<0.01; ***p<0.001.
DOI: https://doi.org/10.7554/eLife.35619.013

The following figure supplement is available for figure 6:

**Figure supplement 1.** Cell surface expression of IL7Rα and Ccr7 are increased in *Rictor* cKO NK cells.
DOI: https://doi.org/10.7554/eLife.35619.014

phosphorylation, modulator protein 14-3-3 binds to FoxO transcription factors and reduces their transcriptional activity by blocking DNA binding and accelerating nuclear exportation (*Brunet et al., 1999*; *Cahill et al., 2001*). Among the FoxO families, FoxO1 is the most abundant one expressed in NK cells and has been shown to negatively regulate the terminal maturation of NK cells by suppressing the transcription of *Tbx21* (*Deng et al., 2015*; *Wang et al., 2016*). Thus, we hypothesized that mTORC2 regulates T-bet expression through the Akt$^{S473}$-FoxO1 axis.

To test this hypothesis, we first evaluated the phosphorylation of Akt and FoxO1. Lack of Rictor resulted in a reduced level of Akt$^{S473}$ phosphorylation in the NK cells from BM (*Figure 6B*, top). Consistent with this, we also detected reduced phosphorylation of FoxO1$^{T24}$ in these *Rictor* cKO NK cells (*Figure 6B*, bottom). These data suggested that the transcriptional activity of FoxO1 is higher in *Rictor* cKO compared to WT NK cells. Indeed, we found a significant enrichment of FoxO1 target genes (p=4.46 × 10–9, Z-score = 1.719) in *Rictor* cKO NK cells compared with WT in the IPA analyses (*Figure 6C*). Ouyang W et al. established a FoxO1-dependent transcriptional program in regulatory T cells (Tregs) through comparing transcriptome among WT, FoxO1 KO and FoxO1 constitutively active Tregs (*Ouyang et al., 2012*). Utilizing FoxO1 target genes described by these authors, we performed gene enrichment analyses of the *Rictor* cKO RNAseq data. Consistent with the IPA analyses, our Fisher's exact test showed enrichment of the FoxO1-target genes with p-value equal to $1.32 \times 10^{-10}$, emphasizing a hyperactive FoxO1 in the *Rictor* cKO NK cells. We further validated several well-established FoxO1 target genes by RT-qPCR using NK cells from both BM and spleen. Consistent with the RNA sequencing data, we found reduced mRNA level of *Tbx21* in CD27$^+$ NK cell subset from both BM and spleen of *Rictor* cKO mice (*Figure 6D*). We also found significantly elevated mRNA level of known FoxO1 activated genes including *Bcl2*, *Klf2*, *Il7r* and *Ccr7* in the CD27$^+$ *Rictor* cKO NK cells compared to the WT (*Figure 6D*) (*Kerdiles et al., 2009*; *Ouyang et al., 2012*). This is also consistent with higher cell surface expression of IL-7Rα and Ccr7 (*Figure 6—figure supplement 1*). Based on these, we conclude that the impaired mTORC2-Akt$^{S473}$-FoxO1 signaling axis results in hyperactive FoxO1 that potentially suppresses the expression of T-bet in *Rictor* cKO NK cells.

## Disruption of mTORC2 does not affect mTORC1 activation

mTORC2 phosphorylates Akt at Serine$^{473}$ and induces maximal kinase activity of Akt (*Guertin et al., 2006*). Given that Akt is an upstream activator of mTORC1, we investigated whether a reduction in Akt kinase activity resulting from mTORC2 disruption affects mTORC1 activation. Being downstream of mTORC1, phosphorylation of rpS6 is nearly abolished in Raptor-deficient NK cells, as expected (*Figure 7A*, left). Important, deficiency of mTORC2 does not perturb mTORC1 signaling as phosphorylation of rpS6 is moderately increased in *Rictor* cKO compared to WT NK cells (*Figure 7A*, right). The moderate augmentation in mTORC1 activity could potentially result from reduced PTEN signaling as indicated by the RNA sequencing data (*Figure 5E*). On the other hand, we did observe a moderate decrease in mTORC2 activity indicated by phosphorylation of Akt$^{S473}$ in *Rptor* cKO NK cells stimulated with IL-15 (*Figure 7B*, left). This is potentially due to reduced expression of IL-15/IL-2 receptor β chain (CD122) as the phosphorylation of STAT5 is also moderately reduced in *Rptor* cKO NK cells (*Figure 4A and B*). As expected, Akt$^{S473}$ phosphorylation was abolished in *Rictor* cKO NK cells (*Figure 7B*, right). Based on these, we conclude that disruption of mTORC2 does not affect IL-15-mediated mTORC1 activation in NK cells.

## Defective anti-tumor response in *Rptor* or *Rictor* cKO mice

To this end, our detailed analyses revealed that both mTORC1 and mTORC2 are critical to the development of NK cells. Next, we asked whether the developmental defects resulting from Raptor or Rictor deficiency might affect physiological response. To address this question, we challenged *Rptor* or *Rictor* cKO mice with B16F10 melanoma cells via tail vein injection, which establishes the lung metastasis tumor model. The critical role of NK cells in anti-tumor immunity has been well established in this model (*Eckelhart et al., 2011*; *Grundy et al., 2007*; *Lakshmikanth et al., 2009*). Compared to corresponding WT control mice, the tumor metastases were much more severe in the lungs of *Rptor* or *Rictor* cKO mice (*Figure 8A − D*). The significant reduction of NK cell number and the terminal mature NK cells in both of the cKO mice might contribute to this defective antitumor response. To further investigate the functional defects in these mice, we conducted *in vivo*

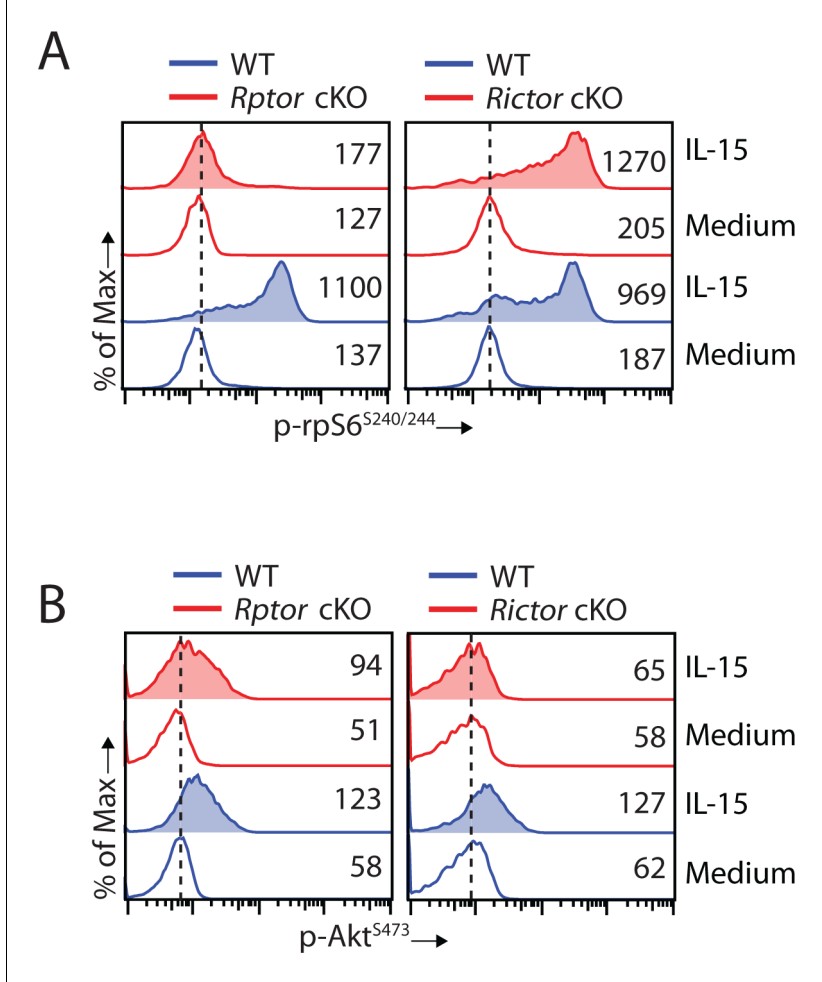

**Figure 7.** Disruption of mTORC2 does not affect mTORC1 activation. (A, B) Splenocytes from WT, *Rptor* cKO (left) or *Rictor* cKO mice (right) were stimulated with either medium or 100 ng/mL IL-15 for 1 hr. Phosphorylation of rpS6$^{S240/244}$ (A) and Akt$^{S473}$ (B) were detected by phosphor-flow and shown as the representative histogram of three independent experiments.

DOI: https://doi.org/10.7554/eLife.35619.015

splenocytes rejection assay using $B2m^{-/-}$ cells, 'missing-self' targets sensitive to NK cells. Similar to the B16F10 tumor challenge, the clearance efficiency of the transferred $\beta_2$-microglobulin-deficient targets cells was significantly impaired in *Rptor* or *Rictor* cKO mice (*Figure 8C and D*). Taken together, these data demonstrated that the NK-cell-mediated anti-tumor response is defective in *Rptor* or *Rictor* cKO mice.

## mTORC1 is essential to the effector functions of NK cells

Next, we evaluate the effector functions of Raptor- or Rictor-deficient NK cells at per cell level. In vivo Poly (I:C)-activating splenocytes were co-cultured with NK cell-sensitive target RMA/s cells or stimulated with IL-12/IL-18 or PMA/Ionomycin. Degranulation (indicated by CD107a expression) and IFN-γ generation were evaluated by flow cytometry. Compared to WT control, Raptor-deficient NK cells had significantly reduced degranulation, and almost abolished IFN-γ production when stimulated by RMA/s cells (*Figure 9A and B*). Consistent with previous findings (*Marçais et al., 2014*), mTORC1 is not required for IL-12/IL-18-mediated IFN-γ production (*Figure 9A and B*). We also observed impaired effector functions of Raptor-deficient NK cells when stimulated with PMA/Ionomycin (*Figure 9A and B*). In contrast, the effector functions of *Rictor* cKO NK cells are intact regardless of the type of stimuli (*Figure 9C and D*). To further evaluate the role of mTORC1 and mTORC2

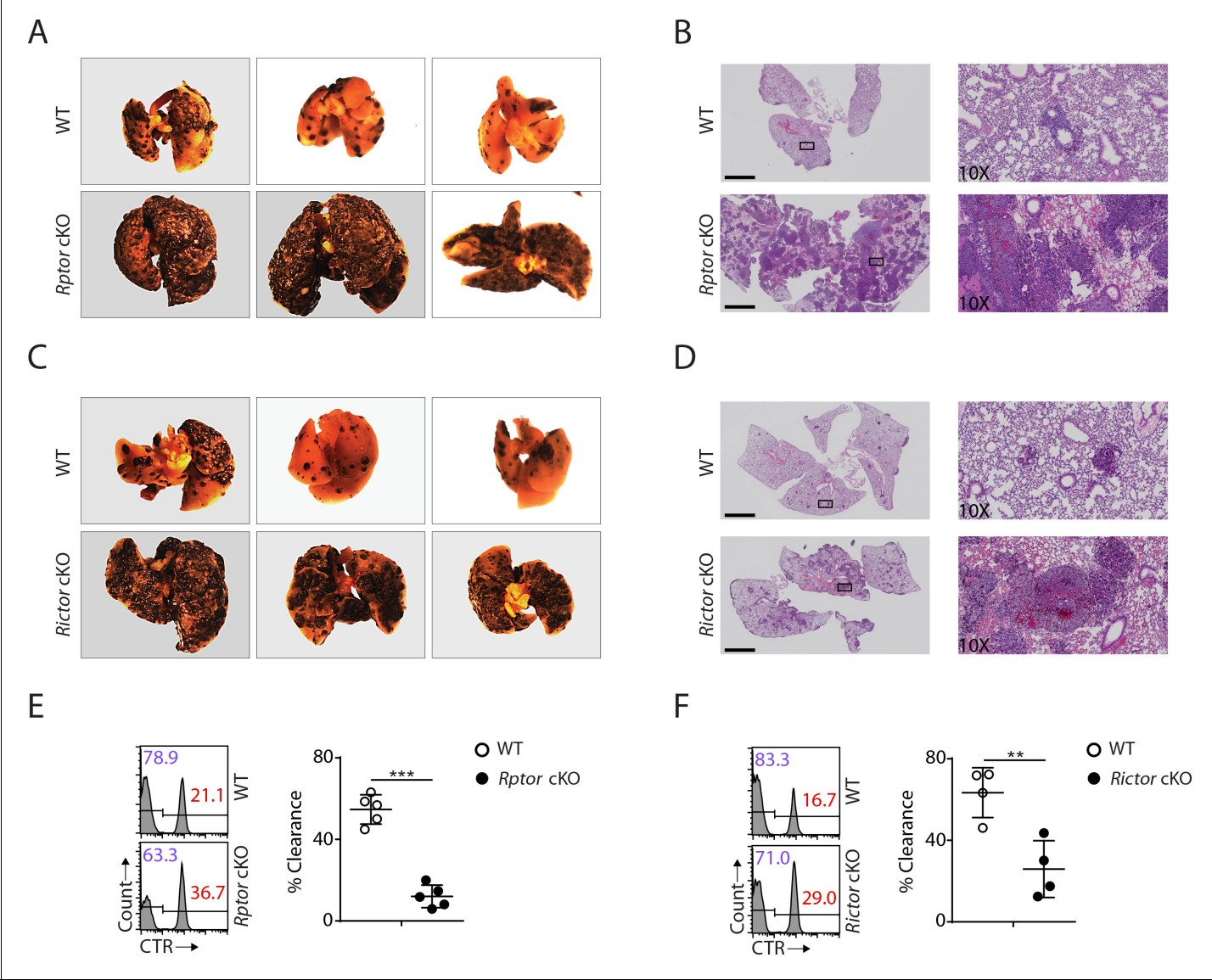

**Figure 8.** Impaired antitumor activity in *Rptor* or *Rictor* cKO mice. (A–D) 200,000 B16F10 tumor cells were intravenously injected into *Rptor* (A, B) or *Rictor* (C, D) cKO mice and their corresponding WT control mice. 14 days post-injection, the lungs were perfused with PBS and harvested for image acquisition (A, C) and HE staining (B, D; Scale bars represent 2.5 mm, and the right side are the 10X exploded view of select regions). The representative images were shown. n = 3–6 for each genotype from two independent experiments. (C, D) CTV-labeled splenocytes from WT C57BL/6 mice were mixed with CTV/CTR-double-labeled splenocytes from $B2m^{-/-}$ C57BL/6 mice at 1:1 ratio. Total $5 \times 10^6$ cells mixed cells were i.v. injected to *Rptor* (E) or *Rictor* (F) cKO mice and their corresponding WT control mice. 18 hr post-injection, the splenocytes from recipient mice were analyzed by flow cytometry. The percentages of WT and $B2m^{-/-}$ lymphocytes were analyzed as the representative histogram (left, gated on CTV+ lymphocytes). The percentage cytotoxicity was also calculated (right). n = 4–5, pooled from two independent experiments. All bar graphs present the mean ± SD. Statistical significance was calculated using unpaired Student t-test. *p<0.05; **p<0.01; ***p<0.001.
DOI: https://doi.org/10.7554/eLife.35619.016

in the effector functions of NK cells, we used IL-2-activated NK cells *in vitro*. Consistent with the critical role of mTORC1 downstream of IL-2 in regulating cell growth and proliferation, the Raptor-deficient NK cells expand poorly in culture. Thus, we used Rapamycin as a surrogate to acutely inhibit mTORC1. Rictor-deficient NK cells respond to IL-2 and expand normally as their WT control, consistent with relatively optimal CD122 expression and STAT5 signaling (*Figure 4C and D*). We used standard [51]Cr-release assay to assess the cytotoxicity potential of NK cells against various target cell lines. Neither inhibition of mTORC1 by Rapamycin nor Rictor deficiency resulted in changes in the

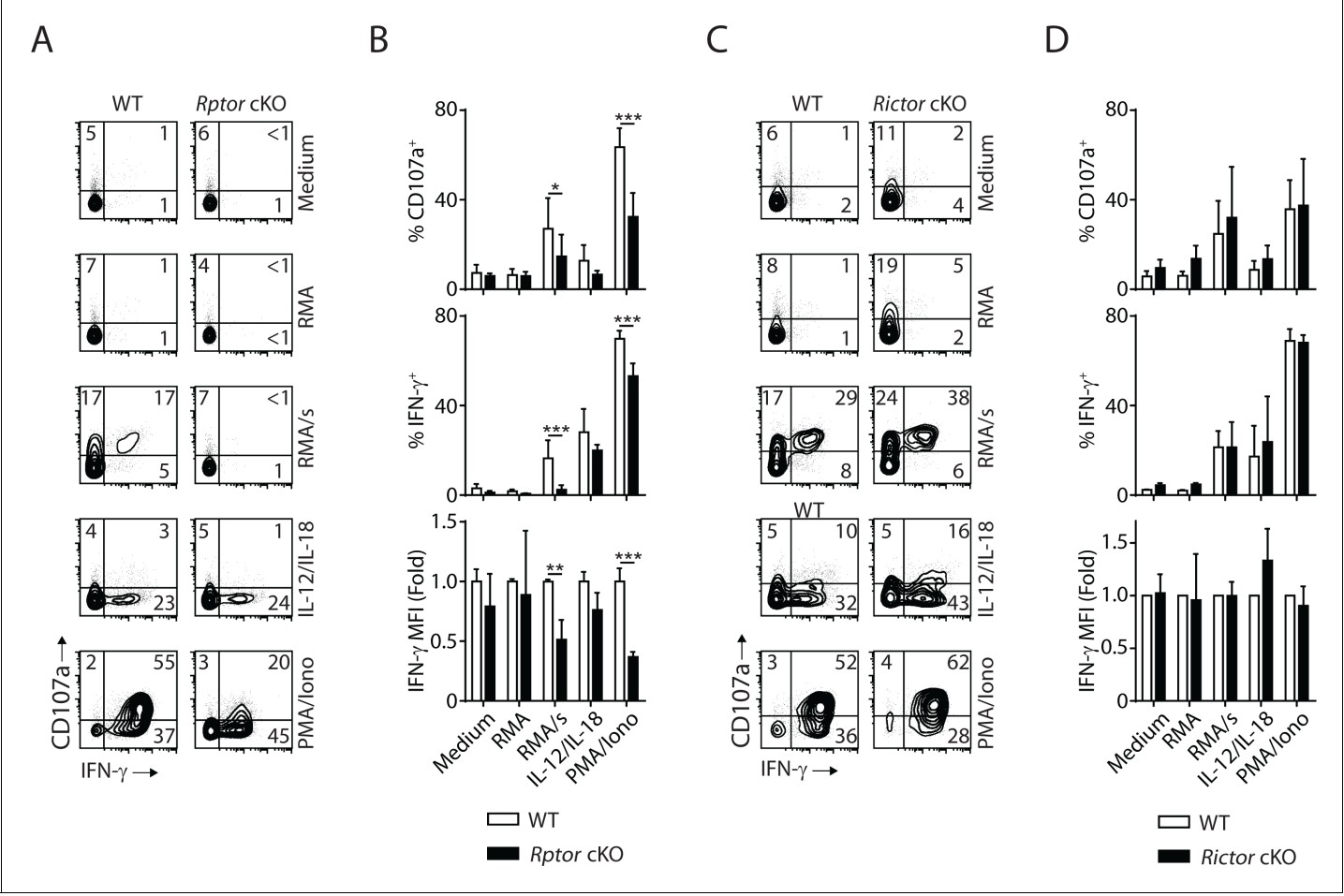

**Figure 9.** mTORC1 is critical for the effector function of NK cells. *In vivo* Poly (**I:C**)-activated splenocytes were co-cultured with RMA/s target cells or stimulated with IL-12/IL-18 or PMA/Ionomycin for 6 hr. Medium and RMA target cells were used as the control groups. Degranulation (indicated by CD107a expression) and IFN-γ generation were assessed by flow cytometry. (**A**) Representative flow plots demonstrating the expression of CD107a and IFN-γ on gated NK cells from *Rptor* cKO and WT mice in different stimuli. (**B**) Percentage CD107a$^+$, IFN-γ+NK cells from *Rptor* cKO and WT mice in different stimuli were quantified. The MFI of IFN-γ was normalized to WT in each condition. n ≥ 5, pooled from four independent experiments. (**C, D**) Expression of CD107a and IFN-γ in NK cells from *Rictor* cKO and WT mice (as shown in **A** and **B**). n = 3, pooled from three independent experiments. All bar graphs present the mean ± SD. Statistical significance was calculated using two-way ANOVA. *p<0.05; **p<0.01; ***p<0.001.
DOI: https://doi.org/10.7554/eLife.35619.017

The following figure supplement is available for figure 9:

**Figure supplement 1.** mTORC1 is essential for the IFN-γ generation downstream of NKG2D.
DOI: https://doi.org/10.7554/eLife.35619.018

ability of NK cells to lyse the sensitive targets (*Figure 9—figure supplement 1A and B*), indicating that both mTOR complexes are dispensable for the conjugate formation and delivery of cytolytic vesicles. The degranulation defects seen in the *in vivo* Poly (I:C)-activated Raptor-deficient NK cells (*Figure 9A and B*) potentially results from the reduced storage of cytolytic granules as mTORC1 is critical in cytokines-mediated priming of NK cells (*Marçais et al., 2014*; *Nandagopal et al., 2014*). As for cytokine generation, Rapamycin inhibits the production of IFN-γ downstream of activating receptor NKG2D, but not PMA/Ionomycin (*Figure 9—figure supplement 1C*). Consistent with the *ex vivo* functional assay (*Figure 9C and D*), IL-2-activated Rictor-deficient NK cells produce IFN-γ comparable to the WT control (*Figure 9—figure supplement 1D*). In summary, these data demonstrate that mTORC1, but not mTORC2, is critical to the effector functions of NK cells.

## Discussion

mTOR plays a critical role in the development and functions of various immune cells (*Powell et al., 2012*; *Weichhart et al., 2015*). In NK cells, the earlier study revealed the essential role of mTOR in controlling development following Ncr1 expression (*Marçais et al., 2014*). This study revealed that lack of mTOR resulted in an impairment in the transition of CD27 SP to DP and defective IL-15-mediated STAT5 activation. , Irrespective of these findings, the independent roles of mTORC1 and mTORC2 remains unknown. Moreover, it is impossible to elucidate the precise molecular mechanisms by which mTOR regulate distinct developmental progression of NK cells with the concurrent loss of both mTORC1 and mTORC2. Therefore, we generated $Ncr1^{iCre}$-mediated conditional deletion of *Rptor* or *Rictor* to disrupt mTORC1 or mTORC2 specifically in NK cells. Phenotypic analyses revealed that both mTOR complexes are critical for the homeostasis of NK cells and that the independent disruption of mTORC1 or mTORC2 results in defective NK cell maturation at distinct developmental stages. The differential alteration of the expression of Eomes and T-bet in *Rptor* or *Rictor* cKO mice provide a potential mechanistic explanation for the maturation defects. In the physiological context, neither *Rptor* nor *Rictor* cKO mice demonstrated robust NK cell-mediated antitumor responses, highlighting the biological consequences of lacking either mTORC1 or mTORC2.

In *Rptor* cKO mice, the homeostatic NK cellularity is disrupted as evidenced by reduced NK cell number in the periphery, reduced steady-state proliferation and impaired migration *in vitro*. Moreover, exclusive loss of mTORC1 significantly impairs NK cell maturation, as demonstrated by accumulation of CD27 SP population and reduced DP and CD11b SP populations. This defect may directly contribute to the accumulated NK cells in the BM, as they gradually obtain migratory capacity following CD11b expression (*Mayol et al., 2011*). Despite our findings using *Rptor* cKO mice, $Ncr1^{Cre}$-mediated deletion of *Pdpk1* or *Tsc1* did not have any impact on NK cell development (*Yang et al., 2016*; *Yang et al., 2015*). This suggests that following *Ncr1* expression, mTORC1 is potentially activated through an alternative mechanism instead of the canonical PI3K-PDK1-Akt-TSC1/2-mTORC1 pathway. In addition, mTORC1 is also likely critical for early NK cell commitment and development at pre-Ncr1 stages based on the phenotype seen in $Pdpk1^{fl/fl}$ $Vav1^{Cre}$ mice (*Yang et al., 2015*). Further work is required to elucidate these up-stream mechanisms that regulate mTORC1 signaling. mTORC1 regulates protein translation through various mechanisms. One of which directly affects the translation of proteins comprising the translational machinery such as eIFs, eEFs, and ribosomal proteins (*Meyuhas, 2000*; *Thoreen et al., 2012*). However, less is known about the regulation of these proteins at the transcriptional level, especially related to the loss of mTORC1. In this context, our RNAseq analyses revealed that selective disruption of mTORC1 leads to increased transcripts level of eIFs, eEFs, and ribosomal proteins, implying an alternative pathway that compensates the protein translation defects. Gene ontology analyses also revealed that oxidative phosphorylation and integrin signaling are impaired in *Rptor* cKO NK cells. These findings are novel and have not been implicated with mTORC1 in lymphocytes. These observations warrant future studies especially related to the effector functions of NK cells.

NK cell maturation is driven by the temporally-regulated production of key transcription factors at distinct developmental stages (*Hesslein and Lanier, 2011*). Our RNAseq analyses revealed that expression of key transcription factors (Nfil3, Id2, Eomes) that govern early NK cell development are impaired in *Rptor* cKO but not in *Rictor* cKO NK cells, correlating with the early developmental defects in *Rptor* cKO mice. Among those three transcription factors, Nfil3 is dispensable for the development of NK cells following *Ncr1* expression (*Firth et al., 2013*). Although both Id2- and Eomes-deficient NK cells have similar developmental phenotype as *Rptor* cKO NK cells (*Delconte et al., 2016*; *Gordon et al., 2012*), we did not detect increased expression of E-protein genes that are suppressed by Id2 in NK cells (*Delconte et al., 2016*). On the other hand, we found significantly reduced expression of Eomes protein and enrichment of Eomes target genes in *Rptor* cKO NK cells. This indicates that among those three transcription factors, the reduced Eomes expression is more likely responsible for the maturation defects due to the exclusive loss of mTORC1.

Compared to mTORC1, less is known about mTORC2, mainly due to the lack of mTORC2-specific inhibitors; however, through using the $Ncr1^{iCre}$ mice model, we are able to study the independent role of mTORC2 specifically in NK cells. Our data reveal that mTORC2 is essential for the development of NK cells. NK cell number is significantly reduced in the periphery of *Rictor* cKO mice, and

this impairment potentially results from reduced steady-state proliferation. Unlike the early maturation impairment at CD27 SP to DP transition by the loss of mTORC1, deletion of *Rictor* causes a defect during the transition from the DP to CD11b SP stage. Importantly, when we assess the expression of T-box transcription factors, Eomes expression is comparable between WT and *Rictor* cKO NK cells, which is consistent with an unaltered transition from CD27 SP to DP stages. This indicates that mTORC1 activation is not affected in Rictor-deficient NK cells during development. The *in vitro* IL-15 stimulation further proves that mTORC2 deficiency did not affect mTORC1 activation despite diminished Akt$^{S473}$ phosphorylation. In contrast, the expression of T-bet is significantly reduced in *Rictor* cKO NK cells, which correlates with the impairment in terminal maturation. Mechanistically, Rictor but not Raptor deficiency results in hyperactive FoxO1 due to abolished mTORC2-Akt$^{S473}$-FoxO1 signaling regulation. FoxO1 has been shown to directly bind the promoter region of T-bet and suppress its expression. However, the upstream signaling pathway that releases the suppression of T-bet from FoxO1 has not been established during NK cell development. RNAseq analyses revealed a hyperactive FoxO1 phenotype in Rictor-deficient NK cells. Reduced steady-state phosphorylation of FoxO1 further emphasizes a more functionally active FoxO1 in the absence of mTORC2 due to abolished Akt$^{S473}$ phosphorylation. Thus, the hyperactive FoxO1 suppresses T-bet expression at transcriptional level in *Rictor* cKO NK cells. While this pathway has previously been shown to control CD8 T cell memory differentiation (*Pollizzi et al., 2015*; *Zhang et al., 2016*), our work establishes that during NK cell development, stimuli (presumably IL-15) activate mTORC2-Akt$^{S473}$-FoxO1 signaling cascade that reveals the suppression of T-bet from FoxO1 and this signaling axis is indispensable to maintain NK cell homeostasis and drive terminal maturation. The RNAseq analyses also revealed that PTEN signaling is impaired in *Rictor* cKO NK cells, which indicates a balanced activation loop consists of PI(3)K-PI(3,4,5)P3-mTORC2-PTEN. This may contribute to the moderately increased mTORC1 activity in *Rictor* cKO NK cells. Future work is warranted to explore this phenomenon. The overall effector functions are normal in Poly (I:C)-primed or IL-2-cultured Rictor-deficient NK cells.

In summary, our study establishes the differential and independent roles of mTORC1 and mTORC2 in the development of NK cells. Our study implies that careful considerations should be taken when utilizing mTOR inhibitors following hematopoietic stem cell transplantation since manipulation of mTOR complexes can potentially compromise NK cell development and repopulation.

## Materials and methods

### Mice, cell lines, and reagents

*Rptor$^{fl/fl}$* and *Rictor$^{fl/fl}$* mice were purchased from the Jackson Laboratory (Bar Harbor, ME). *Ncr1$^{iCre}$* mice were a generous gift from Dr. Eric Vivier (*Narni-Mancinelli et al., 2011*). All mice are in the C57BL/6 background. The *Rptor$^{fl/fl}$* or *Rictor$^{fl/fl}$* mice (WT) were bred with *Ncr1$^{iCre}$* mice to obtain *Rptor$^{fl/fl}$ Ncr1$^{Cre/WT}$* (*Rptor* cKO) or *Rictor$^{fl/fl}$ Ncr1$^{Cre/WT}$* mice (*Rictor* cKO). All mice were maintained in pathogen-free conditions at the Biological Resource Center at the Medical College of Wisconsin. Female and male littermate mice between the ages of 8 to 12 weeks were used. All animal protocols were approved by Institutional Animal Care and Use Committees. The following antibodies and reagents were used in this study. EL4, RMA, RMA/S, and YAC-1 cell lines were purchased from ATCC (Rockville, MD) and maintained in RPMI-1640 medium containing 10% heat-inactivated FBS (Life Technologies, Grand Island, NY). Generation of *H60*-expressing EL4 stable cell lines has been described (*Regunathan et al., 2005*). Authenticity of RMA and RMA/s were tested by the levels of MHC-Class I (H2-K$^b$ and H2-D$^b$). EL4$^{H60}$ and EL4 were validated by the presence of cell-surface H60 protein. YAC-1 was tested by the absence of H-2$^b$ and the presence of H-2$^a$ markers. All these cell lines were regularly tested and are negative for mycoplasma. CD3 (17A2), NK1.1 (PK136), CD49b (DX5), CD27 (LG.7F9), CD11b (M1/70), KLRG1 (2F1), NCR1 (29A1.4), NKG2D (CX5), Ly49H (3D10), NKG2A/C/E (20d5), CD122 (5H4 or TM-b1), Ki-67 (SolA15), Eomes (Dan11mag), T-bet (4B10), CD127 (A7R34), CD244.2 (eBio244F4), CD107a (eBio1D4B), IFN-γ (XMG1.2), Streptavidin-PE, Donkey anti-Rabbit second antibodies are from Thermo-Fisher Scientific (Waltham, MA); Ly49D (4E5), CD45.2 (104), CD132 (TUGm2), CD135 (A2F10), Biotin-Ccr7 (4B12) are from Biolegend (San Diego, CA); Ly49A (A1), Ly49G2 (4D11), Ly49C/I (5E6), p-STAT5$^{Y694}$ (47) are from BD Pharmingen (San Jose, CA); Raptor (24C12), Rictor (53A2), p-Akt$^{S473}$ (D9E), p-rpS6$^{S240/244}$ (D68F8), p-4E-BP1$^{T37/46}$

(236B4), p-FoxO1$^{T24}$/FoxO3a$^{T32}$ are from Cell Signaling Technology (Danvers, MA); β-Actin (ACTBD11B7) is from Santa Cruz Biotechnology (Dallas, TX). Recombinant murine IL-15 is from Peprotech (Rocky Hill, NJ).

## Cell separation, flow cytometry, and cell sorting.

BM cells were flushed, and a single-cell suspension was made by passing through the syringe/needles. Cells from spleen and lymph nodes were prepared by gently grinding the dissected organs with micro slides (VWR, Radnor, PA). Blood was drawn from the cheeks and mixed with 3.8% sodium citrate (Ricca Chemical Company, Batesville, IN). Red blood cells were lysed by RBC lysis buffer (Thermo-Fisher Scientific, Waltham, MA). For liver lymphocytes acquisition, 10 mL PBS was injected into a hepatic artery to perfuse the blood from the liver. After dissecting and grinding the liver, lymphocytes were separate through Percoll (Sigma, St. Louis, MO) gradient centrifugation (40% and 60%). Flow cytometry analyses were conducted in LSR-II (BD Biosciences, San Jose, CA) or MACS-Quant Analyzer 10 (Miltenyi Biotec, Bergisch Gladbach, Germany) and analyzed with FlowJo software (FlowJo LLC, Ashland, OR). For cell sorting, NK cells were first enriched using negative selection kit (STEMCELL Technologies, Vancouver, Canada). The specific subsets of NK cells were further sorted by FACSAria (BD Biosciences, San Jose, CA), and the purity was generally above 95%.

## Intracellular staining and phosphor-flow

Ki-67, Eomes, and T-bet intracellular staining were conducted using Foxp3/Transcription Factor Staining Buffer Set (Thermo-Fisher Scientific, Waltham, MA). For phosphor-flow analysis, BD Phosflow Lyse/Fix Buffer and Perm Buffer III were used (BD Biosciences, San Jose, CA). All procedures were performed following instructions from manufactures.

## Western blotting

Fresh FACS-sorted or IL-2-cultured NK cells were lysed in ice-cold 0.3% CHAPS lysis buffer (25 mM HEPES, pH 7.4; 150 mM NaCl; 1 mM EDTA and 0.3% CHAPS) with phosphatase inhibitor cocktail, PhosSTOP (Roche Diagnostics GmbH, Mannheim, Germany) and proteinase inhibitor cocktail (Sigma, St Louis, MO). Lysates were incubated for 30 min on ice, centrifuged at 15,000 $g$ for 10 min at 4°C. For Western blotting, cell lysates were separated by SDS-PAGE; transferred to PVDF membrane and probed with primary and the secondary Abs conjugated with horseradish peroxidase. The signal was detected by autoradiography films (LabScientific Inc., Livingston, NJ).

## Real-time PCR

Total RNA was extracted from sorted cells using RNeasy Micro Kit (Qiagen, Hilden, Germany). Reverse transcription was conducted using iScript cDNA synthesis kit (Bio-Rad, Hercules, CA). qPCR were performed in Applied Biosystem 7500 (Thermo-Fisher Scientific, Waltham, MA) with SYBR Green-based detection. The transcript levels of β-Actin were used as a control. Primers used for the qPCR reactions in this study were as follows: *Tbx21*-F: 5′-GCCAGGGAACCGCTTATATG-3′, *Tbx21*-R: 5′-GACGATCATCTGGGTCACATTGT-3′; *S1pr5*-F: 5′-GCCTGGTGCCTACTGCTACAG-3′, *S1pr5*-R: 5′-CCTCCGTCGCTGGCTATTTCC-3′; *Bcl2*-F: 5′-CTCGTCGCTACCGTCGTGACTTCG-3′, *Bcl2*-R:5′-CAGATGCCGGTTCAGGTACTCAGTC-3′; *Klf2*-F: 5′-CTCAGCGAGCCTATCTTG-3′, *Klf2*-R: 5′-AGAGGATGAAGTCCAACAC-3′; *Il7r*-F: 5′-GACTACAGAGATGGTGACAG-3′, *Il7r*-R: 5′-GGTGACA TACGCTTCTTCT-3′; *Ccr7*-F: 5′-CCAGCAAGCAGCTCAACATT-3′; *Ccr7*-R: 5′-GCCGATGAAGGCA TACAAGA-3′; *Actb*-F: 5′-GGCTGTATTCCCCTCCATCG-3′, *Actb*-R: 5′-CCAGTTGGTAACAATGCCA TGT-3′.

## RNA sequencing

Total RNA was extracted by Trizol from CD11b$^-$ BM NK cells from littermate WT and *Rptor* cKO mice or CD27$^+$ BM NK cells from littermate WT and *Rictor* cKO mice. (n = 3 per group), followed by poly-A-purification, transcription, and chemically fragmentation using Illumina's TruSeq RNA library kit using the manufacturer's protocol (Illumina, Inc., San Diego, CA). Individual libraries were prepared for each sample, indexed for multiplexing, and then sequenced on an Illumina HiSeq2500. The Trim Galore program (v0.4.1) was used to trim bases with a Phred quality score <20 [https://www.bioinformatics.babraham.ac.uk/projects/trim_galore/] and only reads with a Phred quality score

equal or higher than 20 were taken for analysis. The RSEM program function 'rsem-prepare-reference' (v1.3.0) was used to extract the transcript sequences from the mouse genome (Build GRCm38) [PMID: 21816040] and to generate Bowtie2 indices (Bowtie2 v2.2.8) [PMID: 22388286], followed by read alignment using the 'rsem-calculate-expression' function. Differential expression analysis was performed using the Bioconductor package DESeq2 version 1.12.4 [PMID: 25516281] to compute log2 fold changes and false discovery rate-adjusted p-values. Statistical significance was determined at a false discovery rate threshold of 0.05. Data were analyzed for molecular and functional pathway enrichment using Ingenuity Pathway Analysis (IPA; Qiagen, Redwood City, CA).

## B16F10 lung metastasis model

B16F10 melanoma cells growing in log phase were harvested and resuspended in PBS. $2 \times 10^5$ cells were injected into mice through the tail vein. 14 days post-injection, the recipient mice were sacrificed. The lungs were perfused with 20 mL PBS and dissected for image acquisition.

## *In vivo* splenocytes rejection assay

Splenocytes from WT C57BL/6 mice and $B2m^{-/-}$ C57BL/6 mice were harvested and labeled with Cell Trace Violet (CTV) or CTV plus Cell Trace Red (CTR), respectively. Then, the WT and $B2m^{-/-}$ splenocytes were mixed at 1:1 ratio. The exact percentage of WT and $B2m^{-/-}$ cells (within lymphocytes gate) in the mixture was analyzed by flow cytometry before injection. The ratio of $B2m^{-/-}$/WT before the injection is marked as $R_{pre}$. Total $5 \times 10^6$ cells mixed cells were then retro-orbitally injected into recipient mice. 18 hr post-injection, the splenocytes from recipient mice were analyzed by flow cytometry. The percentages of WT and $B2m^{-/-}$ cells within the lymphocytes gate were acquired. The ratio of $B2m^{-/-}$/WT after injection was marked as $R_{post}$. Percentage clearance was calculated as: % Clearance = $[1-(R_{post}/R_{pre})] \times 100$.

## *In vitro* cytotoxicity assay

Chromium-51 ($^{51}$Cr)-labeled target cells were co-cultured with NK cells at a varied effector to target (E:T) ratios for 4 hr. Percent specific lysis was calculated using amounts of absolute, spontaneous, and experimental $^{51}$Cr-release from target cells.

## *In vitro* functional assay

Poly (I:C) was injected into mice intraperitoneally (10 µg/g). 18 hr later, $2 \times 10^6$ splenocytes were either co-cultured with $2 \times 10^6$ target cells or stimulated with IL-12/IL-18 (10/10 ng/mL), PMA/Ionomycin (50/500 ng/mL) in 24-well plate for 6 hr in the presence of CD107a-PE, Monensin (Thermo-Fisher Scientific, Waltham, MA), and Golgi stop (BD Biosciences, San Jose, CA). For IL-2-cultured NK cells, $1 \times 10^5$ cells were stimulated with plate-bound anti-NKG2D (2 µg/mL) antibody or PMA/Ionomycin in 96-well plate for 6 hr in the presence of Golgi stop for 6 hr. After stimulation, cells were harvested for surface and intracellular IFN-γ staining.

## Statistics

The data were presented as Mean ± SD except for *Figure 4H* which was shown as Mean ± SEM. Statistical analyses were conducted using Prism software (GraphPad, La Jolla, CA). Statistical significance was calculated using unpaired Student t-test or two-way ANOVA for multiple comparisons. The significance is indicated as *p<0.05; **p<0.01; ***p<0.001.

## Acknowledgements

We thank Lucia Sammarco and her Lulu's Lemonade Stand for inspiration, motivation, and support. $Ncr1^{iCre}$ mice were a kind gift from Eric Vivier, Centre d'Immunologie de Marseille-Luminy, France. This research was completed in part with computational resources and technical support provided by the Research Computing Center at the Medical College of Wisconsin. We thank Yongwei Zheng for the help with the tail vein injection. We thank the colleagues from Children's Research Institute Histology Core for the help with the lung sections and HE staining. We are grateful to Zachary Gerbec, Jason Siebert, Nathan Schloemer, and Alex Abel for inputs in the preparation of this manuscript.

## Additional information

### Funding

| Funder | Author |
|---|---|
| National Institute of Allergy and Infectious Diseases | Subramaniam Malarkannan |
| National Cancer Institute | Subramaniam Malarkannan |

The funders had no role in study design, data collection and interpretation, or the decision to submit the work for publication.

### Author contributions

Chao Yang, Conceptualization, Data curation, Software, Formal analysis, Validation, Investigation, Visualization, Methodology, Writing—original draft, Writing—review and editing; Shirng-Wern Tsaih, Data curation, Formal analysis; Angela Lemke, Methodology, Was instrumental in data acquisition as well as analysis and interpretation of results; Michael J Flister, Data curation, Formal analysis, Methodology, Writing—original draft; Monica S Thakar, Funding acquisition, Investigation, Project administration, Writing—review and editing; Subramaniam Malarkannan, Conceptualization, Resources, Data curation, Software, Formal analysis, Supervision, Funding acquisition, Validation, Investigation, Visualization, Methodology, Writing—original draft, Project administration, Writing—review and editing

### Author ORCIDs

Subramaniam Malarkannan http://orcid.org/0000-0002-7511-2731

### Ethics

Animal experimentation: All animal protocols were approved by Institutional Animal Care and Use Committees of the IACUC at the Medical College of Wisconsin, Milwaukee, WI. Medical College of Wisconsin is formally accredited by AAALAC and all the animal care and use-protocols used in this study fully adhere to the specified guide lines of AAALAC. The unique animal protocols that are approved by the IACUC and used in this study are: AUA1500 and AUA1512.

### Decision letter and Author response

Decision letter https://doi.org/10.7554/eLife.35619.024
Author response https://doi.org/10.7554/eLife.35619.025

## Additional files

### Supplementary files

• Supplementary file 1. The expression of Id2 target genes in *Rptor* cKO NK cells.
DOI: https://doi.org/10.7554/eLife.35619.019

• Transparent reporting form
DOI: https://doi.org/10.7554/eLife.35619.020

### Data availability

We have deposited the RNA-Seq data in NCBI SRA "BioSample" database. The SRA BioProject ID is PRJNA434424.

The following dataset was generated:

| Author(s) | Year | Dataset title | Dataset URL | Database, license, and accessibility information |
|---|---|---|---|---|
| Malarkannan S | 2018 | RNA-Seq data from | http://www.ncbi.nlm.nih.gov/bioproject/?term=PRJNA434424 | Publicly available at NCBI BioProject (Accession no: |

PRJNA434424)

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
