## [Decision Letter]

Thank you for submitting your article "mTORC1 and mTORC2 differentially regulate natural killer cell development" for consideration by *eLife*. Your article has been reviewed by two peer reviewers, and the evaluation has been overseen by a Reviewing Editor and Michel Nussenzweig as the Senior Editor. The following individual involved in review of your submission has agreed to reveal her identity: Megan Cooper (Reviewer #1).

The reviewers have discussed the reviews with one another and the Reviewing Editor has drafted this decision to help you prepare a revised submission.

The paper was reviewed by two scientists expert in NK cell biology and a member of the Board of Reviewing Editors (BRE). They endeavored to provide a consensus statement and together suggest the authors submit a revised manuscript describing the in vitro functions of Rictor and Raptor ko NK cells as suggested by both reviewers. In addition, the reviewers wondered about the novelty of the findings, given that they were almost exactly as expected, based on the previous findings regarding mTor and NK cells. A revised manuscript should help readers understand why the current manuscript is an advance even though the findings were predictable. FoxO1 deserves more prominence in the Discussion as one reviewer felt that this finding was the more novel findings for NK cells. The full reviews are given below. The authors should endeavor to address all comments in a revised manuscript.

Reviewer #1:

This manuscript by Yang et al. investigate the requirements for mTORC1 and mTORC2 on NK cell development and homeostasis using a genetic model. Overall, the manuscript presents novel findings and provides significant insight into mechanisms regulating NK cell development. The experiments are well-controlled and the manuscript is well-written. The authors use an NKp46-specific iCre mouse to target Rptor and Rictor, primary components of the mTORC1 and mTORC2 complexes respectively. They performed a phenotypic analysis of the NK compartment and demonstrate a block in early NK cell differentiation with Rptor ko and a block in the final stages of NK differentiation with Rictor ko. Analysis of TFs known to be important for NK differentiation reveal Eomes (Rptor) and Tbet (Rictor) as downregulated, as well as differences in CD122 expression and IL-15 signaling. There is some data suggesting impaired BM egress and trafficking including an in vivo assay and decreased mRNA expression of S1PR5, although this is not that developed and it isn't clear whether there is truly a trafficking defect. The focus, and strength of the manuscript, is the investigation of the molecular pathways altered by deletion of Rptor and Rictor in NK cells, including analysis of Eomes and Tbet and the identification of hyperactive FoxO1 activity.

There are no in vitro functional assays of the NK cells presented (cytokine production, degranulation, killing). The authors do present in vivo data with reduced cell clearance. However, it is expected that lower NK cell numbers would lead to lower in vivo target clearance (and susceptibility to a tumor model), while in vitro studies would reveal any functional differences on a single cell level.

Reviewer #2:

The authors specifically delete Raptor and Rictor in NK cells and observe slight differences (between the 2 KO mice) in the impact of deletion on different stages of NK cell maturation. Overall, peripheral NK cell numbers drop 2-3 fold in each KO mouse, nicely demonstrating non-redundancy, and each KO mouse not surprisingly susceptible to B16 tumors. Overall this study is quite descriptive, and relatively incremental given the prior findings demonstrating the importance of mTOR signaling in NK cells.

The NK cells from KO mice could be analyzed in greater depth. For instance, do the KO NK cells kill and secrete cytokines similar to WT on a per cell basis? These standard ex vivo functional assays need to be performed – and using littermate controls. Does enhanced IL-7R expression (and loss of Eomes) actually mean that ILC1 development is favored over NK cells (and may explain the lack of trafficking because they are now tissue resident)? What might this mean in the context of nutrient sensing during development?

The authors implicate an AKT-FoxO1 pathway in dysregulated gene expression in NK cells when Rictor is ablated, but much of this mechanistic data is very correlative. ChIP studies would validate the authors' claims of direct regulation of the implicated genes by FoxO1 instead of T-bet (as this transcription factor is increased in Rictor deficient cells).

[Editors' note: further revisions were requested prior to acceptance, as described below.]

Thank you for submitting your article "Raptor and Rictor differentially promote natural killer cell development" for consideration by *eLife*. Your article has been reviewed by a Reviewing Editor and Michel Nussenzweig as the Senior Editor.

The revised manuscript does not address one of the major concerns cited in the consensus statement. Specifically, mentioned previously was – "The reviewers wondered about the novelty of the findings, given that they were almost exactly as expected, based on the previous findings regarding mTor and NK cells. A revised manuscript should help readers understand why the current manuscript is an advance even though the findings were predictable."

In the current revised manuscript, it is not clear if this was addressed. We invite you to re-submit another revision that deals with aforementioned criticism. For example, you should better outline the limit of knowledge in the area, not just for NK cells. You should discuss (in the Discussion) what could have been predicted by prior studies, what was not, and discuss why your paper is more than confirmation of predictions. Your work on FoxO1 could be highlighted here. A rebuttal letter outlining these changes should accompany the revised manuscript. Please also outline all changes made in the revised manuscript with respect to the original submission and reviews.

---

## [Author Response]

Reviewer #1:[…] There are no in vitro functional assays of the NK cells presented (cytokine production, degranulation, killing). The authors do present in vivo data with reduced cell clearance. However, it is expected that lower NK cell numbers would lead to lower in vivo target clearance (and susceptibility to a tumor model), while in vitro studies would reveal any functional differences on a single cell level.

We analyzed the functional capabilities of these NK cells using two different systems.

The first system is in vivo Poly (I:C)-activated NK cells stimulated with NK cell-sensitive target cell line RMA/s, IL-12/18 or PMA/Ionomycin. We included the medium and RMA cell line which is not NK cell sensitive target as the controls. We evaluated the degranulation and IFN-γ generation using flow cytometry. The data demonstrated that Raptor-deficient NK cells have impaired effector functions when stimulated with RMA/s cell lines or PMA/Ionomycin. Consistent with the previous report, mTORC1 seems to be independent for IL-12/18-mediated activation (Figure 9A and B). On the contrary, Rictor-deficient NK cells do not have any functional defects regardless of the type of stimuli (Figure 9C and D).

The second system we used is the IL-2-cultured NK cells. Specifically, NK cells from spleen are expanded in vitro and subject to functional assay. Due to the pool expansion of Raptor-deficient NK cells, we used Rapamycin to acutely inhibit mTORC1 as the surrogate. The cytolytic potential of NK cells is not affected either by Rapamycin (Figure 9—figure supplement 1A) or Rictor deficiency (Figure 9—figure supplement 1B), indicating that neither mTOR complex is involved in degranulation progress. As for cytokine generation, mTORC1, but not mTORC2, is required for IFN-γ production downstream of activating receptor NKG2D (Figure 9—figure supplement 1C and D).

In summary, our data indicate that mTORC1 is essential for the effector function of NK cells, and mTORC2 is not critical for these processes. These additional data are included in the revised manuscript (Figure 9 and Figure 9—figure supplement 1).

Reviewer #2:[…] The NK cells from KO mice could be analyzed in greater depth. For instance, do the KO NK cells kill and secrete cytokines similar to WT on a per cell basis? These standard ex vivo functional assays need to be performed – and using littermate controls. Does enhanced IL-7R expression (and loss of Eomes) actually mean that ILC1 development is favored over NK cells (and may explain the lack of trafficking because they are now tissue resident)? What might this mean in the context of nutrient sensing during development?

Throughout this study, we have used the respective littermate controls (*Rptor^fl/fl^*and *Rictor^fl/fl^*).

As for the standard functional assay, please see the response to the specific comments from reviewer #1.

As for the enhanced IL-7R expression, it was seen in Rictor-deficient NK cells which do not have a defect in Eomes expression (Figure 3C). We do not think that ILC1 development is favored over NK cells in the *Rictor* cKO mice due to the fact that T-bet expression is significantly reduced, which implies defects in ILC1 development (Figure 3D).

The authors implicate an AKT-FoxO1 pathway in dysregulated gene expression in NK cells when Rictor is ablated, but much of this mechanistic data is very correlative. ChIP studies would validate the authors' claims of direct regulation of the implicated genes by FoxO1 instead of T-bet (as this transcription factor is increased in Rictor deficient cells).

We agree that ChIP assay will further strengthen our argument on the suppression of T-bet expression by hyperactive FoxO1 in Rictor-deficient NK cells. However, ChIP experiment requires a large number of cells. With the developmental defects in the cKO mice, it is technically challenging for us to conduct the ChIP assay.

To clarify the misunderstanding, the expression of T-bet is reduced in both mRNA and protein level in NK cells from *Rictor* cKO mice.

[Editors' note: further revisions were requested prior to acceptance, as described below.]

The revised manuscript does not address one of the major concerns cited in the consensus statement. Specifically, mentioned previously was – "The reviewers wondered about the novelty of the findings, given that they were almost exactly as expected, based on the previous findings regarding mTor and NK cells. A revised manuscript should help readers understand why the current manuscript is an advance even though the findings were predictable."In the current revised manuscript, it is not clear if this was addressed. We invite you to re-submit another revision that deals with aforementioned criticism. For example, you should better outline the limit of knowledge in the area, not just for NK cells. You should discuss (in the Discussion) what could have been predicted by prior studies, what was not, and discuss why your paper is more than confirmation of predictions. Your work on FoxO1 could be highlighted here. A rebuttal letter outlining these changes should accompany the revised manuscript. Please also outline all changes made in the revised manuscript with respect to the original submission and reviews.

As suggested by the Editors, we have addressed the major concerns related to

a) ‘novelty of the findings, given that they were almost exactly as expected, based on the previous findings regarding mTor and NK cells’ and

b) ‘why the current manuscript is an advance even though the findings were predictable’. Please see our rebuttal addressed below:

“outline the limit of knowledge in the area, not just for NK cells”. We have added additional information that describes our limited knowledge related to the signaling pathways that regulate lymphocyte development (Introduction, third paragraph).

“outline the limit of knowledge in the area, not just for NK cells”. One of the major unknown is the molecular interplay between mTORC1 and mTORC2 and its functional relevance. We have articulated this in the Introduction, fourth paragraph.

“outline the limit of knowledge in the area, not just for NK cells”. We have added emphasis on how our study is an advancement over the previous findings (Introduction, fifth paragraph).

“why the current manuscript is an advance”. We have emphasized why independently investigating mTORC1, and mTORC2 is a critical advancement compared to the earlier study using mTOR KO mice (Discussion, first paragraph).

“why the current manuscript is an advance even though the findings were predictable”. We summarize our novel findings related to the oxidative phosphorylation and integrin signaling that were impaired in Raptor-deficient NK cells, which were not predicted or discussed by any earlier studies (Discussion, third paragraph).

“why the current manuscript is an advance even though the findings were predictable”. We highlight our finding that lack of Rictor did not affect the activation of mTORC1 during NK cell development. This is another critical finding that can neither be predicted nor defined using mTOR-deficient mice (Discussion, fifth paragraph).

“Your work on FoxO1 could be highlighted here”. We have highlighted our findings that Rictor-deficient but not Raptor-deficient mice exhibit a hyperactive FoxO1 phenotype in NK cells. Through our study, we have established a novel pathway, mTORC2-AktS473-FoxO1-T-bet in developing NK cells (Discussion, fifth paragraph).